



# Arctic sea ice cover data from spaceborne SAR by deep learning

Yi-Ran Wang and Xiao-Ming Li

Key Laboratory of Digital Earth Science, Aerospace Information Research Institute, Chinese Academy of Sciences, Beijing, 100094, China.

*Correspondence to*: Xiao-Ming Li (lixm@radi.ac.cn)

**Abstract.** Widely used sea ice concentration and sea ice cover in polar regions are derived mainly from spaceborne microwave radiometer and scatterometer data, and the typical spatial resolution of these products ranges from several to dozens of kilometers. Due to dramatic changes in polar sea ice, high-resolution sea ice cover data are drawing increasing attention for polar navigation, environmental research, and offshore operations. In this paper, we focused on developing an approach for

deriving a high-resolution sea ice cover product for the Arctic using Sentinel-1 (S1) dual-polarization (horizontal-horizontal, HH, and horizontal-vertical, HV) data in extra wide swath (EW) mode. The approach for discriminating sea ice from open water by synthetic aperture radar (SAR) data is based on a modified U-Net architecture, a deep learning network. By employing an integrated stacking model to combine multiple U-Net classifiers with diverse specializations, sea ice segmentation is achieved with superior accuracy over any individual classifier. We applied the proposed approach to over 28,000 S1 EW

images acquired in 2019 to obtain sea ice cover products in a high spatial resolution of 400 m. By converting the S1-derived sea ice cover to concentration and then compared with Advanced Microwave Scanning Radiometer 2 (AMSR2) sea ice concentration data, showing an average absolute difference of 5.55 % with seasonal fluctuations. A direct comparison with Interactive Multisensor Snow and Ice Mapping System (IMS) daily sea ice cover data achieves an average accuracy of 93.98 %. These results show that the developed S1-derived sea ice cover results are comparable to the AMSR and IMS data in terms

of overall accuracy but superior to these data in presenting detailed sea ice cover information, particularly in the marginal ice zone (MIZ). Data are available at: http://www.dx.doi.org/10.11922/sciencedb.00273 (Wang and Li, 2020)

## 1 Introduction

Sea ice retreat, particularly in the Arctic, has been one of the most significant responses to global climate change (Serreze and Barry, 2011). Therefore, sea ice cover and sea ice concentration are vital parameters for conducting climate change research,

navigation in polar regions, and the success of offshore operations.

Spaceborne microwave radiometers have provided the longest time series of sea ice concentration data in polar regions. The large-scale recording of sea ice began with the advent of the Nimbus-5 Electrically Scanning Microwave Radiometer (ESMR) in 1972, followed by the launch of dual-polarization multifrequency systems such as the Nimbus-7 Scanning Multichannel Microwave Radiometer (SMMR) (Gloersen et al., 1993) in 1978. Subsequently, the Special Sensor Microwave Imager (SSM/I)

multichannel radiometer system (Hollinger et al., 1990) on board the Defense Meteorological Satellite Program (DMSP)





satellites and their successor, the Special Sensor Microwave Imager and Sounder (SSMIS) (Kunkee et al., 2008), provided long time-series records of sea ice concentration from 1987 to 2016. The typical sea ice concentration data provided by SSM/I and SSMIS have a spatial resolution of approximately 25 km (Comiso et al., 1997; Parkinson et al., 1999). A new generation of passive microwave sensors have been launched in recent decades, namely, is represented by the NASA Aqua Advanced

Microwave Scanning Radiometer for the Earth Observing System (AMSR-E), which launched in May 2002, followed by the Advanced Microwave Scanning Radiometer 2 (AMSR2) on board the JAXA Global Change Observation Mission-Water (GCOM-W) satellite that launched in May 2012. AMSR-E daily average sea ice concentration products are derived mainly from 19 GHz and 37 GHz channels with the bootstrap technique (Comiso and Sullivan, 1986) with a resolution of 12.5 km. AMSR2 provides two types of sea ice concentration data: one is based on the bootstrap algorithm (Comiso and Sullivan, 1986)

and utilizes vertically polarized brightness temperatures measured from 19 and 37 GHz channels with a resolution of 12.5 km, while the other is based mainly on the ARTIST Sea Ice (ASI) algorithm (Spreen et al., 2008, 2005) and employs information from a higher frequency of 89 GHz, resulting in an increased spatial resolution of 6.25 km. To date, operationally available sea ice concentration data retrieved by spaceborne radiometers have provided reliable measurements of sea ice variations in polar regions. For instance, research shows that the Arctic sea ice extent has decreased since 1978 (Stroeve et al., 2005) and

that warming in the Arctic will continue with a rate greater than the current global average (Pachauri et al., 2014).

Due to their large spatial coverage and polar orbits, the microwave radiometers mentioned above are able to acquire daily sea ice concentration information over the Arctic and Antarctic. However, the typical spatial resolution of sea ice data from a spaceborne microwave radiometer ranges from a few to tens of kilometers. Arctic sea ice is experiencing an ongoing rapid decline (Onarheim et al., 2018) and is consequently becoming younger (Nghiem et al., 2007) and thinner (Kwok and Rothrock,

2009) and is drifting faster (Rampal et al., 2009). On the other hand, while the retreat of sea ice, the spatial and temporal variations of the Arctic marginal ice zone (MIZ) draws increasing attention (Strong and Rigor, 2013), where the significant interaction between sea ice and ocean dynamics might be an important feedback to sea ice retreat (Thomson and Rogers, 2014). Hence, to better understand sea ice dynamics and its interaction with ocean dynamics (particularly in the MIZ) at different spatial and temporal scales, remote sensing-based sea ice information with a high spatial resolution is more desirable than ever.

Spaceborne synthetic aperture radar (SAR) has proven to be an ideal remote sensing technique for generating detailed sea ice information because of its inherent capability to image the surface at a high resolution (up to one meter to date) independent of sunlight and weather conditions. Moreover, its polarimetric capability enables SAR to have different responses to sea ice types and open water. Since the first civilian SAR instrument, Seasat, was launched in 1978, sea ice monitoring in polar regions has become a primary task of operating spaceborne SAR satellites. Previous studies on sea ice monitoring by spaceborne SAR

have focused mainly on the discrimination of sea ice and open water (Hong and Yang, 2018;Komarov and Buehner, 2017), the classification of sea ice types (Wang et al., 2018;Boulze et al., 2020), the detection of icebergs (Power et al., 2001), and the investigation of sea ice drift (Frost et al., 2017). The present study concerns the extraction of sea ice by spaceborne SAR; accordingly, a brief summary of the state-of-the-art SAR-related segmentation techniques for sea ice and open water is given below.



Among the existing SAR-based sea ice segmentation approaches, it is widely agreed that observations acquired under cross-polarization (horizontal-vertical, HV, or vertical-horizontal, VH) are more effective than those retrieved under co-polarization (horizontal-horizontal, HH, or vertical-vertical, VV) because the former is less sensitive to sea surface backscatter (Dierking, 2013;Scheuchl et al., 2004). However, the strong contrast between sea ice and open water in cross-polarization can be limiting for thin ice with a smooth surface or for open water under strong winds. The difference between co-polarization and cross-

polarization data has proven to be an optimal combination for distinguishing between sea ice and open water (Karvonen, 2013; Tan et al., 2018). Recently proposed sea ice segmentation approaches are based mainly on traditional machine learning methods. First, texture features, such as the gray-level cooccurrence matrix (GLCM), energy, correlation, dissimilarity, and entropy, are manually selected and extracted and then fed into traditional machine learning algorithms such as a support vector machine (SVM) (Zakhvatkina et al., 2017;Leigh et al., 2013;Liu et al., 2014;Li et al., 2020), a random forest algorithm (Tan

et al., 2018), or an artificial neural network (Ressel et al., 2015). The performance of traditional machine learning depends heavily on the selection of features. Moreover, although various studies have demonstrated that texture features, such as those based on the GLCM, can effectively reflect the discrepancies between sea ice and open water patterns (Soh and Tsatsoulis, 1999;Clausi and Zhao, 2003;Clausi, 2001), these features fail in cases where sea ice and open water have similar patterns in SAR images, such as an SAR image presenting both windy sea and thin ice surfaces.

The difficulty in designing suitable features for segmentation stems from the complex nature of weather conditions and sea ice states. Hence, manually exhausting all useful texture features for distinguishing sea ice and open water in different situations is challenging. Convolutional neural network (CNN) is a good way to solve this problem. Rather than using hand-crafted features, the CNN input is the original data (image), and a CNN can automatically and hierarchically learn the features in each network layer. Therefore, the sea ice segmentation model proposed in this study is built upon a particularly successful CNN

called U-Net (Ronneberger et al., 2015), which has been widely used in image segmentation and has achieved competitive performance. The novel character of the U-Net architecture provides opportunities for deriving pixelwise sea ice segmentation results by high-resolution spaceborne SAR data and with a relatively small number of training samples.

Although various algorithms and methods have been developed for sea ice segmentation by spaceborne SAR data, it seems that valuable SAR-derived sea ice information has not been widely exploited compared with radiometer-retrieved sea ice

concentration data, which are routinely utilized for sea ice monitoring in polar regions. This situation may be attributed to two aspects. On the one hand, the existing methodologies have limitations and have seldom been validated with a large dataset. On the other hand, spaceborne SAR data acquisitions over polar regions are often discontinuous due to the limitations of onboard storage or other requested tasks. Alternatively, Sentinel-1A and Sentinel-1B (S1A and S1B, respectively) compose a spaceborne SAR constellation that boasts a significantly shortened revisit frequency of less than one day at high latitudes, e.g.,

polar regions. Moreover, S1A and S1B have broadly acquired extensive amounts of data in polar regions since their constellation was formed: more than approximately 2500 SAR images in extra wide swath (EW) mode are acquired each month by these two satellites in the Arctic. Therefore, S1A and S1B provide a unique chance to generate high-resolution sea ice information with a high coverage frequency in polar regions.
In this study, we focused on developing a method for deriving high-resolution Arctic sea ice cover information from S1A and
S1B based on a deep learning architecture, namely, U-Net. Moreover, aiming for a future operational service, we attempted to
generate a standard SAR-derived sea ice cover product by using a full year of S1 images acquired in the Arctic. The remainder
of this paper is organized as follows. The datasets used for developing and validating the algorithm are briefly described in
Section II. Section III presents the proposed sea ice segmentation approach based on the U-Net architecture. Following the
development of the proposed algorithm, comparisons among SAR-derived sea ice cover data, AMSR2 sea ice concentration
data and Interactive Multisensor Snow and Ice Mapping System (IMS) daily sea ice cover data are presented in Section IV.
Section V describes the generated S1-derived Arctic sea ice cover product based on the proposed approach. A discussion and
the conclusions are presented in the last two sections.

## 2 Datasets

### 2.1 Sentinel-1 data

To effectively cover the vast Arctic region, S1A and S1B extensively acquire SAR data in EW mode, which has a swath width
of approximately 400 km. Moreover, the EW mode data in the Arctic are acquired in dual-polarization (HH and HV) and are
particularly suitable for sea ice monitoring. The S1 EW images used are in ground range detected, medium resolution (GRDM)
format with a pixel size of 40 m × 40 m. The incidence angle of the EW data varies from 18.9° in the near range to 47.0° in
the far range. More than 28,000 S1 EW images in dual polarization were collected over the Arctic region (72.5° N-83° N)
during 2019. The processing scheme applied to the S1 EW images is described in detail in subsection 3.1.

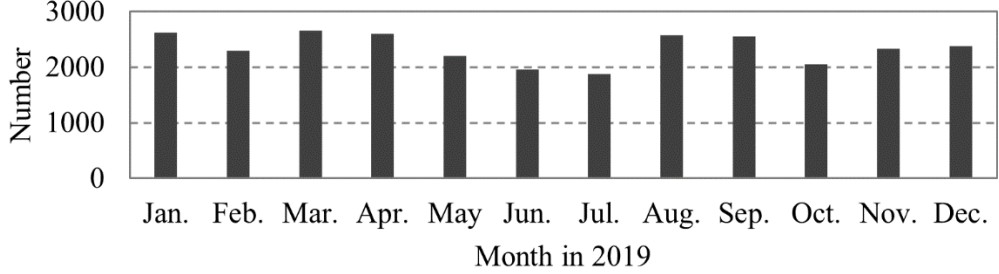

**Figure 1. Number of S1 EW images collected in the Arctic region (72.5° N-83° N) in each month in 2019.**

### 2.2 AMSR2 sea ice concentration data

The daily AMSR2 sea ice concentration product with a spatial resolution of 6.25 km released by the University of Bremen was
used for a comparison with the S1-derived Arctic sea ice cover data. The AMSR2 sea ice concentration data are retrieved
mainly based on the ASI algorithm (Spreen et al., 2008, 2005), which contains an empirical model to retrieve the sea ice

concentration and a statistical model of the atmospheric influence. The ASI algorithm mainly uses the difference between the brightness temperatures at 89 GHz under V and H polarizations. The 89 GHz channel has the highest resolution among all the

channels of the AMSR2 instrument but is influenced more greatly by the atmosphere (e.g., water vapor and cloud liquid water). Thus, the bootstrap algorithm (Comiso and Sullivan, 1986) is employed in conjunction, as it uses the 19 and 37 GHz channels and is therefore less sensitive to atmospheric phenomena (but has a coarser resolution). Accordingly, the S1-derived sea ice cover data were first converted into sea ice concentration data on a regular grid of 6.25 km and then matched with the AMSR2 sea ice concentration data on a pixel-by-pixel basis.

**2.3 IMS sea ice cover data**

Another reference dataset, the sea ice cover data from the IMS (https://nsidc.org/data/G02156/versions/1) released by the National Snow and Ice Data Center (NSIDC), was employed for a comparison with the S1-derived Arctic sea ice cover data. The data are considered valid at 0:00 UTC each day. To determine the presence of sea ice, visible imagery is first retained when not obstructed by clouds; then, passive microwave data and the National Ice Center (NIC) weekly sea ice analysis product

are applied depending on the time of year, resolution and data latency (Ramsay, 1998). In this study, IMS data at a spatial resolution of 1 km are used. As both sources of data record sea ice cover information, the IMS data and S1-derived data were compared directly in corresponding pixels.

**3 Methodology**

**3.1 Overall S1-based sea ice segmentation approach**

The overall architecture of the proposed approach for deriving sea ice cover from S1 EW images in HV and HH polarization is depicted in Figure 2. First, the S1 EW images are preprocessed, which includes radiometric calibration, denoising, incidence angle correction, resampling, and land masking. Then, the preprocessed HV- and HH-polarized data are synthesized into an RGB false-color composite, which serves as the input to the proposed sea ice segmentation model. During the training process, a small proportion of the RGB false-color composites is labeled as training data and fed into the U-Net segmentation model,

generating well-trained sea ice classifiers. The integrated stacking model combines the classifiers from the U-Net model to generate an aggregate sea ice classifier. The aggregate classifier is then applied to all other unlabeled data to generate Arctic sea ice cover data.



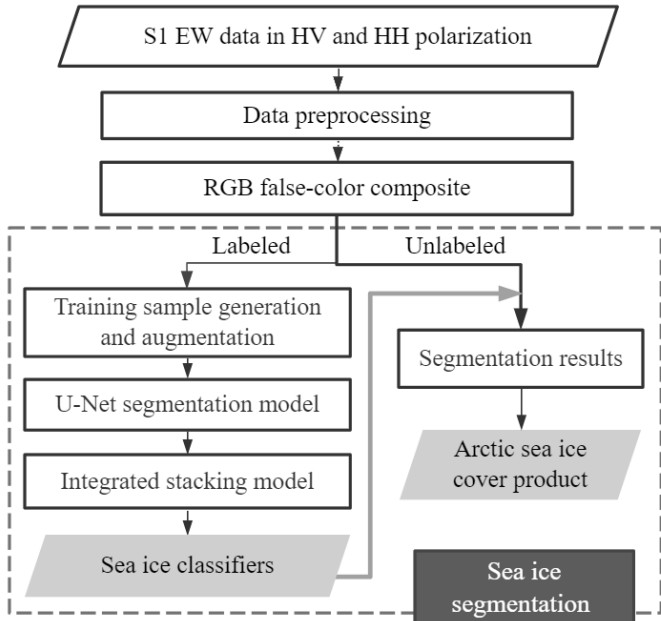

**Figure 2. Flowchart of the proposed method for deriving sea ice cover information from S1 EW images in HV and HH polarization.**

**3.2 S1 image preprocessing**

S1A and S1B EW data in HV and HH polarization were used to develop the proposed algorithm and generate the Arctic sea ice cover product.

The radar backscatter of S1 EW images in HH polarization changes rapidly with variation in the incidence angle. Thus, prior to using these data to derive sea ice cover information, all the EW data in HH polarization are processed for an incidence angle

correction. The linear regression method for the HH backscatter versus elevation angle introduced in (Murashkin et al., 2018) is used. Figure 3 (a) and (b) shows an example of an S1 EW HH-polarized image before and after an incidence angle correction. In contrast, the HV-polarized image does not reveal a significant sensitivity to the incidence angle; therefore, the incidence angle correction is not applied.

The S1 EW images in HV polarization are strongly affected by the scalloping effect and thermal noise. Thus, all the EW data

in HV polarization are denoised using the method proposed in (Li, 2020). This denoising method improved upon previous methods (Park et al., 2017) by segmenting the image into more azimuthal blocks and introducing a variance factor to discriminate homogeneous and inhomogeneous blocks, thereby deriving a more accurate scaling factor and balancing factor. In addition, we proposed a new method in (Li, 2020) for eliminating residual noise (i.e., multiplicative noise) at the sub swath boundaries of EW data. Figure 3 (c) and (d) shows an example of an S1 EW HV-polarized image before and after denoising,

demonstrating good performance in removing both additive and multiplicative noise present in the EW HV-polarized image.



As the original S1 EW images are quite large, these data are downsampled to achieve a more manageable file size to be handled by the sea ice segmentation model. The denoised and calibrated S1 images are averaged by a $10 \times 10$ window, resulting in a change in pixel size from 40 m to 400 m. Thus, the pixel-based sea ice segmentation results also have a pixel size of 400 m. The land area is masked by the Global Self-consistent Hierarchical High-resolution Geography Database (GSHHG,

https://www.ngdc.noaa.gov/mg g/shorelines/gshhs.html) with the full grid resolution of 1×1 arc-minute.

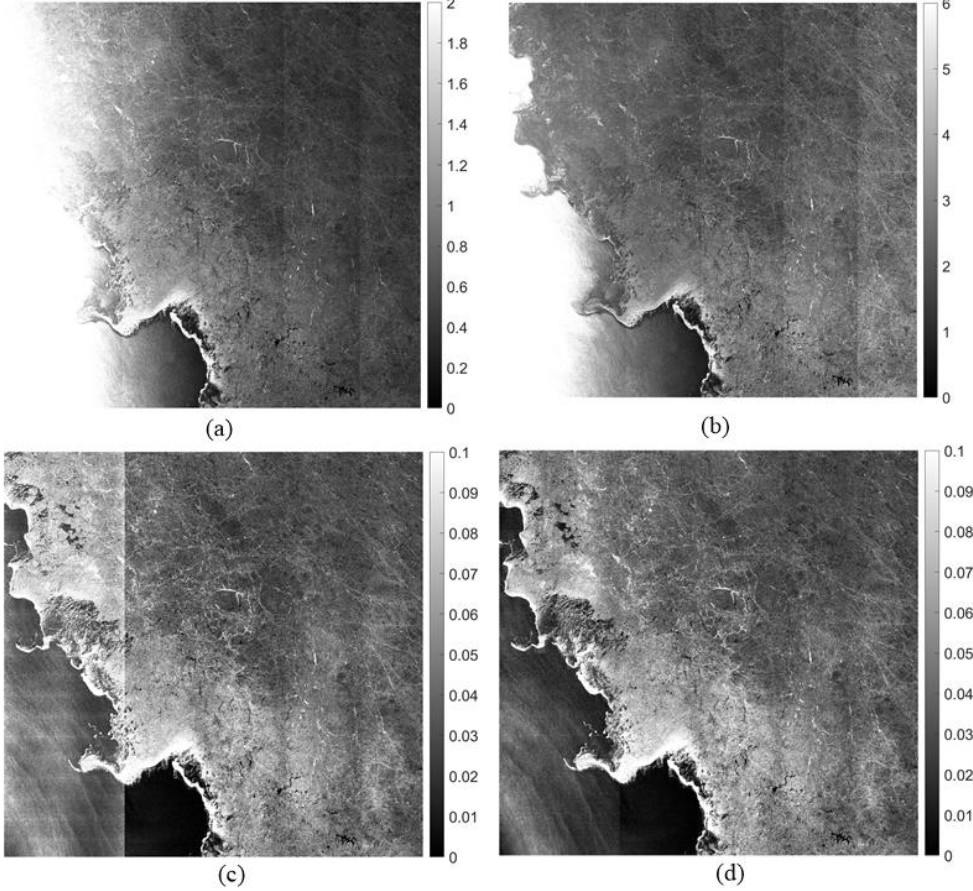

**Figure 3. Example of the denoising process. (a) The original S1 EW HH-polarized image. (b) S1 EW HH-polarized image after an incidence angle correction. (c) The original S1 EW HV-polarized image. (d) Denoised S1 EW HV-polarized image using the denoising method (Li, 2020). (Image ID: S1B_EW_GRDM_1SDH_20180711T073522_20180711T073622_011760_015A2A_4965)**

**3.3 Combination of S1 co- and cross-polarization data**

For sea ice segmentation by spaceborne SAR data, the radar backscatter intensity naturally constitutes the basis of the determination. Many studies consider the cross-polarization channel to be very effective for sea ice detection because it is sensitive to ice-induced volume scattering, while the sea surface generally presents surface scattering. Thus, in the cross-polarization channel, the radar backscatter of sea ice is generally higher than that of open water. However, this is not always

the case. Under strong winds, open water can present radar backscatter intensities comparable to those of sea ice (an example

is shown in Figure 4 (b), where the sea surface wind speed varies between approximately 8 m/s and 18 m/s according to the ERA5 reanalysis wind field at synoptic time (Hersbach et al., 2020)). Moreover, sea ice with a smooth surface can have a low backscatter intensity (Aldenhoff et al., 2019) (see Figure 4 (d) for example). These substantial variations in the SAR radar backscatter intensities of sea ice and open water make sea ice segmentation a challenging task. As mentioned in the

introduction, the difference between co-polarization and cross-polarization data on sea ice and open water is also effective for distinguishing between sea ice and open water. Therefore, in addition to incorporating cross-polarization data directly, we adopted polarization ratio (HH/HV) and polarization difference (HH-HV) data in the proposed approach.

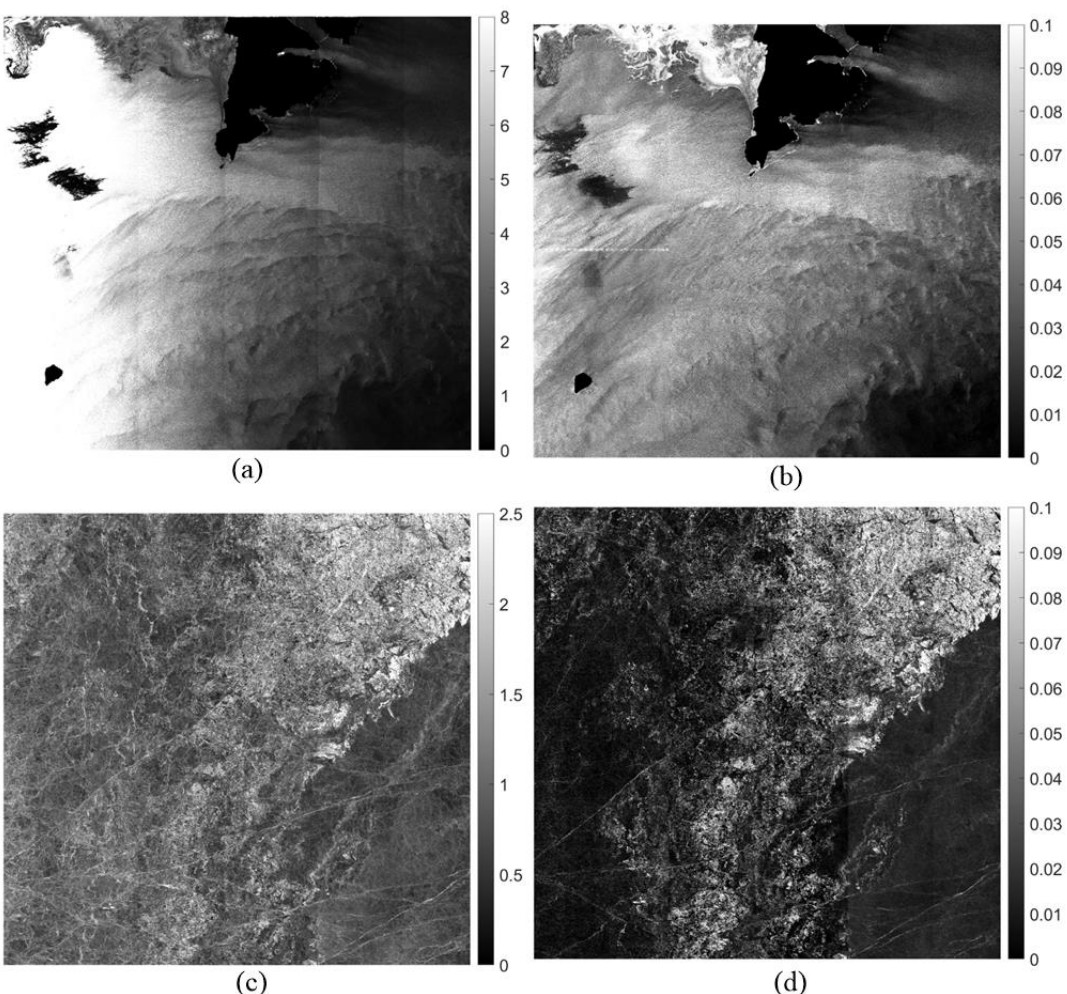

**Figure 4. Examples of S1 EW images presenting the challenges in sea ice segmentation. (a) HH-polarized and (b) HV-polarized S1**
**EW images of a windy sea surface. (c) HH-polarized and (d) HV-polarized S1 EW images of a smooth thin ice surface. (Image ID:**
**(a)    and    (b)    S1A_EW_GRDM_1SDH_20190130T060740_20190130T060840_025703_02DB20_85BC; (c    and    (d)**
**S1B_EW_GRDM_1SDH_20190115T194015_20190115T194115_014509_01B066_11F1)**

Figure 5 illustrates the process of combining the HV-polarized data with the polarization ratio and polarization difference data into an RGB false-color composite, which serves as the input in the sea ice segmentation process. For the HV-polarized S1

images, the radar backscatter in linear units is scaled to 0-255 after discarding 2 % of the maximum and minimum values each. Then, each image is scaled according to its data range to maximize the texture features. The polarization ratio and polarization difference data are first converted into decibel units and then stretched to fixed thresholds of [2 dB, 7 dB] and [-2 dB, 3.5 dB], respectively, to keep the absolute difference between the HV- and HH-polarized data. These thresholds were determined according to the statistics of approximately 200 S1 EW images in different seasons. The polarization ratio or polarization

difference values beyond these two ranges are replaced by the corresponding thresholds, and then a linear stretch to 0-255 is applied. Finally, the scaled HV-polarized data, polarization difference data, and polarization ratio data serve as the red, green, and blue channels, respectively, to synthesize an RGB false-color composite image. The RGB false-color composite clearly presents more textures and details than either the HV-polarized image or the HH-polarized image and therefore lays a better foundation for further sea ice segmentation tasks.

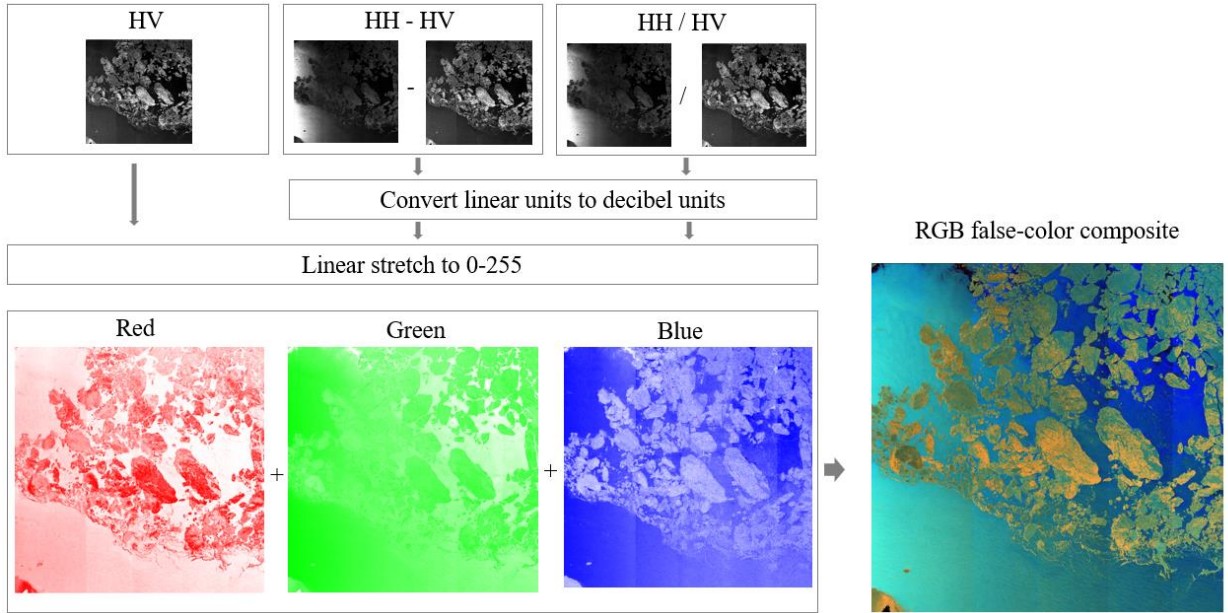


**Figure 5. Illustration of combining HV-polarized data with polarization ratio and polarization difference data into an RGB false-color composite, which presents more textures and details than either HV- or HH-polarized images.**

### 3.4 Generation of training samples

Precisely labeled samples are necessary for the successful development of machine learning algorithms. We first

discriminated sea ice and open water by using 251 S1 EW images based on our previously developed SVM classification (Li et al., 2020), from which the good results (judged by visual inspection) were further used as initial training data. Then, manual checking was performed to improve the correctness and completeness of the classification results. The selection of the training data considered both the data acquisition location (see Figure 6 (a)) and the data acquisition season (see Figure 6 (b)) to represent diverse sea ice conditions. Figure 6 (c) shows examples of four RGB false-color composite images and their

corresponding precisely labeled data, where sea ice is labeled 1 (white) and open water is labeled 0 (black).

We took two-thirds of the total 251 labeled S1 EW images as the training dataset, and we used the remaining one-third of the samples as the evaluation dataset. The labeled dataset is further grouped for the integrated stacking model, which will be described in a later section. To fit the network, first, the labeled data are cropped into patches with dimensions of $256 \times 256$ pixels. Then, training sample augmentation is performed for each patch by rotating and flipping the image, and the expanded

dataset is further processed by adjusting the hue (as illustrated in Figure 6 (d)) to generate additional samples. The augmentation process dramatically increases the quantity and diversity of the training samples, which improves the performance of the deep learning model and reduces overfitting. The evaluation dataset is also cropped, but augmentation is not performed.

**Figure 6. (a) Spatial distribution of the S1 images used for training the sea ice segmentation model. The images are shown as RGB false-color composites, and the land is masked in gray. (b) Histogram of the seasonal distribution of the training data. (c) Examples**

of data labeling: the top row shows four RGB false-color composite images, and the corresponding labeled images are in the bottom row. (d) Illustration of the augmentation process to increase the quantity and diversity of the samples in the training dataset.

## 3.5 Modified U-Net

The overall architecture of the proposed sea ice segmentation model is shown in Figure 7. U-Net is particularly good at sea ice segmentation in SAR images for the following two reasons. 1) The network can perform localization effectively to provide high-resolution segmentation masks by labeling each pixel of the input image with a corresponding class, i.e., sea ice or open water. 2) The network works well with small datasets and is relatively robust against overfitting, although obtaining a large number of labeled samples from remote sensing data is challenging.

The architecture of the sea ice segmentation model contains two parts, i.e., an encoder part and a decoder part. The encoder, which captures the discriminative features in the image, is composed of a stack of convolutional layers, batch norm layers, and rectified linear unit (ReLU) operations, followed by a max pooling layer, where each max pooling layer reduces the spatial resolution of the input layers by a factor of 2. Then, the batch norm layers are added to each of the blocks to increase the learning speed. The decoder part, which semantically projects the discriminative features learned by the encoder onto the pixel

level, consists of upsampling and concatenation operations, followed by convolutional layers. To obtain pixel-based segmentation results, at every decoder step, we used skip connections by concatenating the output of the transposed convolutional layers with the feature maps from the encoder at the same level. High-resolution features from the contracting layers are combined with the upsampled output; hence, these layers increase the resolution of the output. A final convolutional operation with a kernel size of (1, 1) and a sigmoid activation function is performed at the output side of the network. The

network outputs the ultimate segmentation mask in terms of a $256 \times 256$ matrix (same size as the input) with values between 0 and 1. In the segmentation mask, the closer a pixel value is to 1, the more likely it is sea ice, and vice versa for water.

The network is trained using Adam optimization (Kingma and Ba, 2014) with a batch size of 8. A specialized segmentation loss function that combines binary cross-entropy and dice loss is chosen to evaluate the final results as

$$H_p(q) = -\frac{1}{N}\sum_{i=1}^{N} y_i \log\big(p(y_i)\big) + (1 - y_i)\log\big(1 - p(y_i)\big) \tag{1}$$

where $y$ is the label (1 for sea ice and 0 for water) and $p(y)$ is the predicted probability of the pixel being sea ice for all $N$ pixels in each image. A perfect model would have a performance of $H_p(q) = 0$.

The input RGB false-color composite is cropped into patches of $256 \times 256$ pixels and then fed into the model. Predictions made for these patches by the sea ice segmentation model are regrouped accordingly. Neighboring patches have an overlap of 12 pixels to reduce artifacts along the edges of the patches.




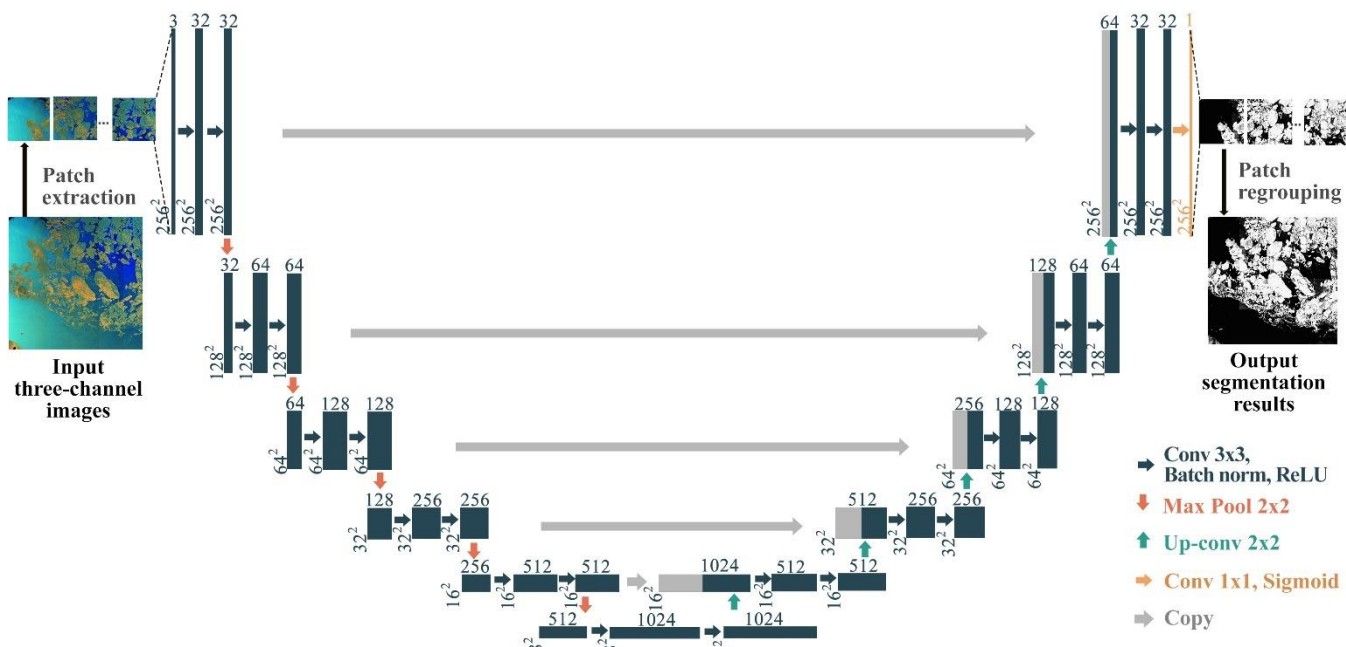

**Figure 7. The overall architecture of the proposed sea ice segmentation model for the S1 EW data based on the U-Net deep learning algorithm.**

## 3.6 Integrated stacking model

During the training and validation of the U-Net model, it is almost impossible to find a perfect model for all scenarios. For example, one model specializes in detecting continuous and large areas of sea ice but tends to output biased segmentation results for windy sea surfaces (e.g., examples shown in Figure 4 (a), (b)), while another model performs well at segmenting regions with highly mixed sea ice and open water (particularly over the MIZ) but performs poorly at containing thin ice with a smooth surface (e.g., examples shown in Figure 4 (c), (d)). This is understandable, as Wolpert and Macready (Wolpert and

Macready, 1997) demonstrated in their "no free lunch" theorem, because an algorithm that performs well in one class of problems must "pay" for that accuracy with degraded performance on a set of all remaining issues.

The performance of each model diverges upon being fed a different set of training data. We divided the training samples into groups and trained the U-Net model with different datasets, thereby generating several classifiers. In our sea ice segmentation approach, we adopted the idea of stacked generalization (Wolpert, 1992) to enable each model to fully utilize its strengths and

mitigate its weaknesses, resulting in higher accuracy and sensitivity. Stacked generalization, or "stacking", is an ensemble machine learning algorithm that involves combining the outputs of several networks into an aggregate output, which often improves the accuracy over any individual output.

As illustrated in Figure 8, the architecture of the proposed integrated stacking model consists of two levels. The first level, i.e., level 0, is formed by base classifiers. Five U-Net classifiers with diverse specializations are selected as the base classifiers of




level 0. Figure 9 shows several cases to give a more visual representation of the selected classifiers and the integrated stacking model. The first two columns contain S1 images in HV polarization and the corresponding RGB false-color composites. The following five columns are the outputs from the selected five classifiers with diverse specializations presented in grayscale with white for 1 and black for 0: the closer the number is to 1, the more likely it is sea ice, and vice versa for water. Notably, models 2 and 3 are specialized for large areas of sea ice, especially newly formed sea ice, whereas models 1 and 4 produce

fewer wrong segmentation results for high wind sea surfaces, and models 3 and 5 deliver more details when ice floes are mixed with water.

The five classifiers are then applied to all the labeled datasets. The output segmentation results, together with the four RGB false-color composite images, are then used as the input data to level 1. We found that the performance of each classifier in level 0 is highly correlated with the percentage of sea ice in one S1 EW image; thus, the sea ice proportion is calculated and

added to the inputs to level 1. The level-1 model is a neural network containing 10 hidden layers that combines the outputs of the base models. Note that level 0 and level 1 use the same training set; however, the dataset in level 0 is divided into several subsets for a different classifier, but in level-1, all the data are fed into the neural network. For the final output of the integrated stacking model, a value of 0.5 is taken as the threshold to judge whether a pixel is classified as 0 (open water) or 1 (sea ice), as shown in the last column of Figure 9, with yellow representing sea ice, cyan representing open water, and gray representing

land.

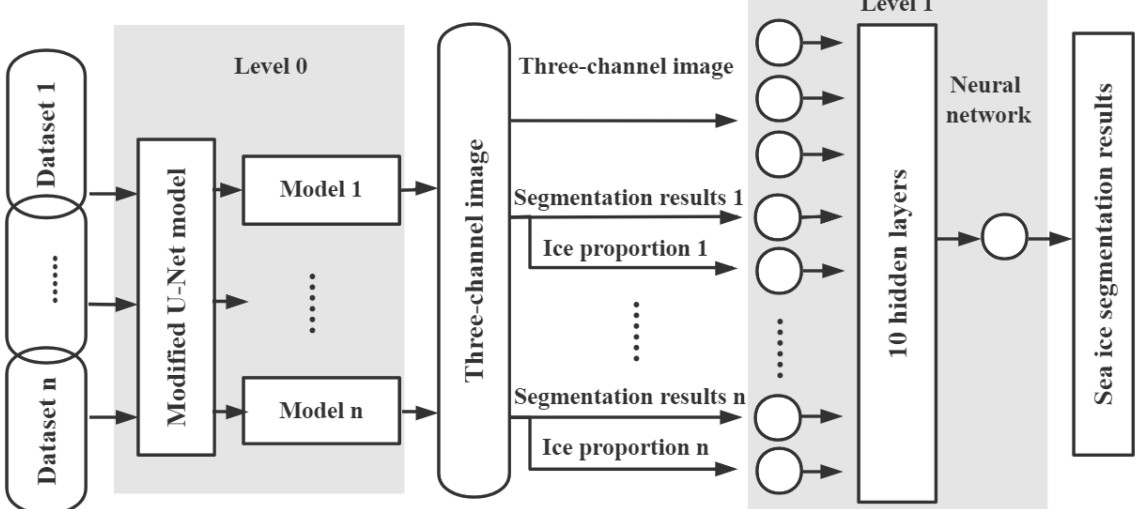

**Figure 8. Overall architecture of the proposed integrated stacking model for generating the S1-derived sea ice cover data.**



| HV polarization | Three-channel image | Results from model 1 | Results from model 2 | Results from model 3 | Results from model 4 | Results from model 5 | Final output |

**Figure 9. Five cases illustrating the combination of selected U-Net classifiers into the integrated stacking model. The first two columns are S1 images in HV polarization and the corresponding input RGB false-color composites. The following five columns are the outputs from the selected five classifiers with diverse specializations, and the last column is the final output of sea ice cover (yellow and cyan represent sea ice and open water, respectively).**

## 4 Results

We applied the developed U-Net-based sea ice segmentation model to over 28,000 S1 EW images acquired in the Arctic in 2019 to obtain sea ice cover data at a spatial resolution of 400 m. Those S1-derived sea ice cover data are compared with AMSR2 sea ice concentration data and IMS sea ice cover data.

### 4.1 Comparison with the AMSR2 sea ice concentration data

We conducted a comparison between the S1-derived Arctic sea ice cover data and the AMSR2 sea ice concentration data. The

S1-derived sea ice cover data, with a spatial resolution of 400 m, were converted into sea ice concentration on a regular grid of 6.25 km, the same as the spatial resolution of the AMSR2 data. Then, the sea ice concentration data were matched with the AMSR2 data on a pixel-by-pixel basis. Figure 10 shows an example of the S1-derived sea ice cover, the corresponding sea ice concentration, and a subsequent comparison with the AMSR2 data. This image was acquired from the Fram Strait and included large areas of both open water and floating sea ice. Figure 10 (a) shows the RGB false-color composite image, and Figure 10

(b) is the corresponding sea ice cover result. Figure 10 (c) presents the calculated sea ice concentration based on the S1-derived sea ice cover data, while the spatially collocated AMSR2 sea ice concentration data are shown in Figure 10 (d). Figure 10 (e) shows the differences in the sea ice concentration between AMSR2 and S1: red indicates the area where the S1-derived sea ice concentration is lower than the AMSR2 result, while blue indicates the opposite. Within the ice water mixing area in the northwest, the AMSR2 sea ice concentrations are overestimated, while in the ice water junction area, the concentrations are

underestimated; this is the main reason for the absolute difference of 11.03 % in this example.

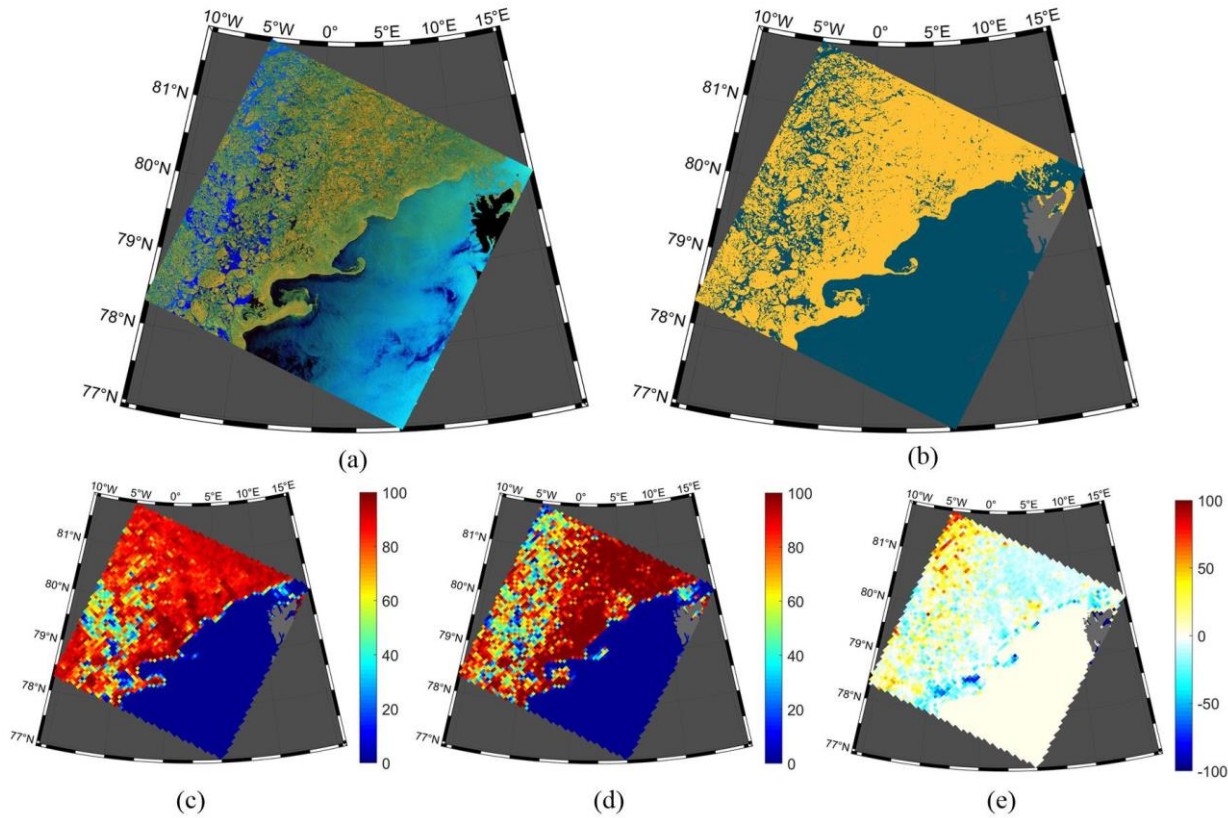

**Figure 10. Example of S1-derived sea ice cover data and its comparison with the AMSR2 data. (a) RGB false-color composite image. (b) S1-derived sea ice cover result. Yellow indicates sea ice, and cyan indicates open water. (c) Sea ice concentration based on the S1-derived result. (d) AMSR2 sea ice concentration. (e) Differences in the sea ice concentration between the AMSR2 data and S1-**
**derived results (AMSR2 - S1). (Image ID: S1A_EW_GRDM_1SDH_20190601T072854_20190601T072958_027483_0319D7_E0F1)**





For a more quantitative analysis, the daily absolute differences between the S1-derived and AMSR2 sea ice concentration data are plotted in Figure 11, together with the daily average AMSR2 sea ice concentration of the region covered by S1 data. The solid lines show the 7-day moving average results. During the summer season (June, July, and August), the sea ice concentration drops from 80 % to approximately 40 %, and the absolute difference shows a noticeable increase to

approximately 10 %; during the other seasons, the absolute difference is almost ubiquitously less than 6 %. For the whole year of 2019, the average absolute difference between the Arctic sea ice product derived from over 28,000 S1 images and the AMSR2 data is 5.55 %. Research has demonstrated that sea ice concentration estimates from passive microwave observations are typically inaccurate in Arctic summer due to the similar microwave radiation characteristics of sea ice and open water, which is mainly attributed to atmospheric effects (Han et al., 2018). Accordingly, the error of the AMSR2 sea ice concentration

data was estimated based on comparisons with 1) in situ ice observations, 2) ice concentration retrievals using other microwave algorithms, and 3) ice concentration data derived from higher-resolution optical sensors. The AMSR2 data show low errors at moderate and high ice concentrations (above 65 %), where the error should not exceed 10 %. However, at low ice concentrations, the accuracy is low: the absolute error is 25 % at 0 % ice concentration and decreases for higher ice concentrations (Spreen et al., 2008). These errors are a plausible reason for the relatively large difference between the S1-

derived sea ice concentration and AMSR2 data in the summer season, particularly in the period from middle July to the beginning of August.

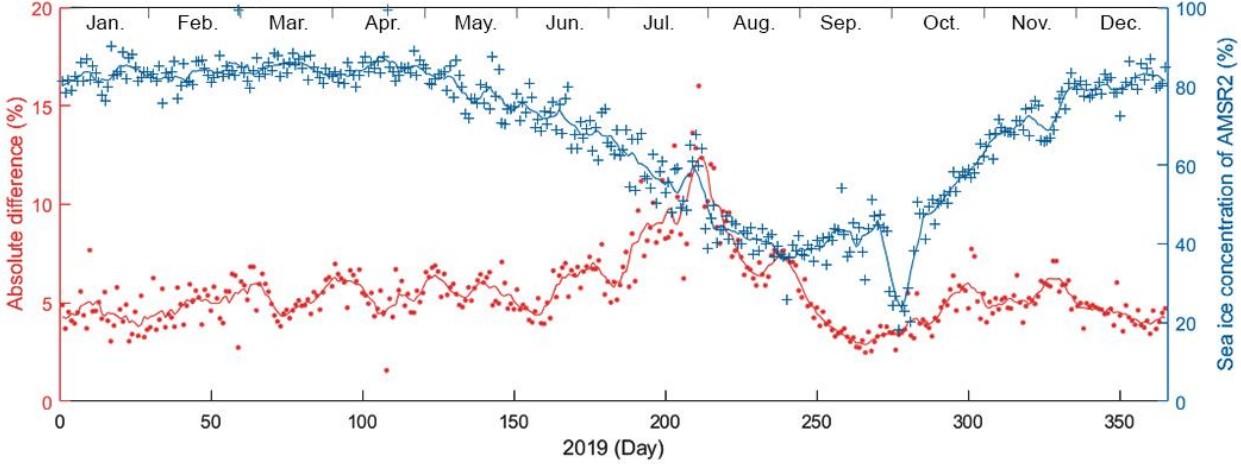

**Figure 11. Comparison between the S1-derived Arctic sea ice concentration data and AMSR2 data for the whole year of 2019. The red dots reflect the absolute daily difference, and the red line is the 7-day average absolute difference. The blue crosses are the daily**

**sea ice concentrations in the S1-covered area calculated based on the AMSR2 data, and the blue line is the 7-day average sea ice concentration.**

## 4.2 Comparison with the IMS sea ice cover data

We also conducted a similar comparison between the S1-derived Arctic sea ice cover data and the IMS sea ice cover data. As the 400 m pixel size of the S1-derived sea ice cover data is comparable to that of the IMS data on a grid size of 1 km, we



directly compared the IMS data and S1-derived results in corresponding pixels. Figure 12 shows an example of the S1-derived

sea ice cover and its comparison with the IMS data in a vital area of the Northeast Passage in the Laptev Sea. Figure 12 (a)

shows the RGB false-color composite image of this example, and Figure 12 (b) and Figure 12 (c) show the S1-derived sea ice

cover and IMS data, respectively. Figure 12 (d) shows the differences between the IMS and S1 results. The comparison is

quantitatively evaluated by the accuracy, i.e., $Accuracy = (n_{TP} + n_{TN}) * 100\ \%/n_{total}$, where the corresponding classified

sea ice and open water pixels are recorded as true positive ($n_{TP}$) and true negative ($n_{TN}$), respectively, and $n_{total}$ denotes the

total pixels of the derived result. As Figure 12 shows, the S1-derived sea ice cover result is more detailed, as the S1-derived

outcome is acquired on a pixel-by-pixel basis, while the IMS data are smoothed, leading to the relatively low accuracy of 81.78

% for this example.

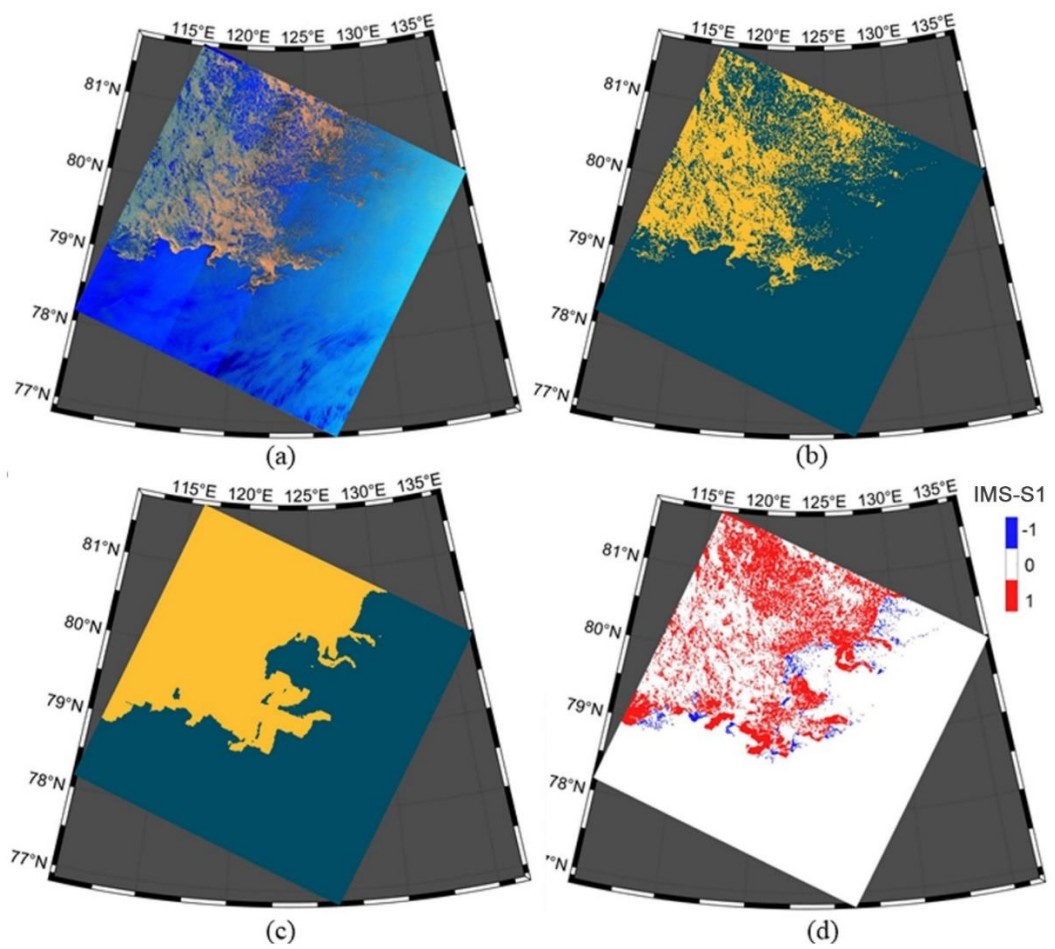

**Figure 12. Example of S1-derived sea ice cover and its comparison with the IMS data. (a) RGB false-color composite image. (b) and (c) S1-derived sea ice cover and IMS sea ice cover, with yellow indicating sea ice and cyan indicating open water. (d) Differences in sea ice cover between the IMS and S1-derived results (IMS - S1). (Image ID: S1B_EW_GRDM_1SDH_20190822T232250_20190822T232354_017705_0214F8_D76F)**



The daily accuracy of the S1-derived sea ice cover compared with the IMS data is plotted in Figure 13. For the whole year of
2019, the overall accuracy of the S1-derived Arctic sea ice data based on more than 28,000 images is 93.98 % compared with
the IMS data. From January to April, the accuracy is approximately 95 % with only minor fluctuations. From May onwards,
the accuracy starts to decrease and reaches the lowest accuracy of 88 % (according to the 7-day average result) at the end of
July before quickly increasing to 95 % again through August. From September until the end of the year, the accuracy is
approximately 95 % with relatively large fluctuations. The sea ice proportion (plotted in blue in Figure 13) representing the
percentage of the daily sea ice area in the S1-covered region calculated based on the IMS data. Before July, the accuracy varies
with a similar trend to the variation in the sea ice proportion. At the end of July, the sea ice proportion increases dramatically
from approximately 60 % to 70 %, and the corresponding comparison shows a minimum accuracy. Up to the end of September,
although the sea ice proportion continues to decrease, the comparison suggests that the accuracy significantly increases. The
variation in the accuracy is similar to the variation in the absolute difference achieved in comparison with the AMSR2 data.
The discrepancy between the IMS data and the S1-derived results may be attributed to three aspects: (1) The IMS data are
produced using data from various satellites with different spatial resolutions, and this may lead to smoother results in the data
fusion process, whereas the S1-derived sea ice cover is pixel-based. This is especially obvious in areas with highly mixed sea
ice and open water (e.g., the example shown in Figure 13). (2) The daily IMS data are compiled based on various satellite
observations within a day, whereas the S1-derived results are snapshots of the sea ice conditions at the time of SAR data
acquisition. The temporal variation of sea ice can also lead to some differences. (3) Some error is caused by the limitations of
the algorithm when deriving sea ice cover based on S1 and IMS data.

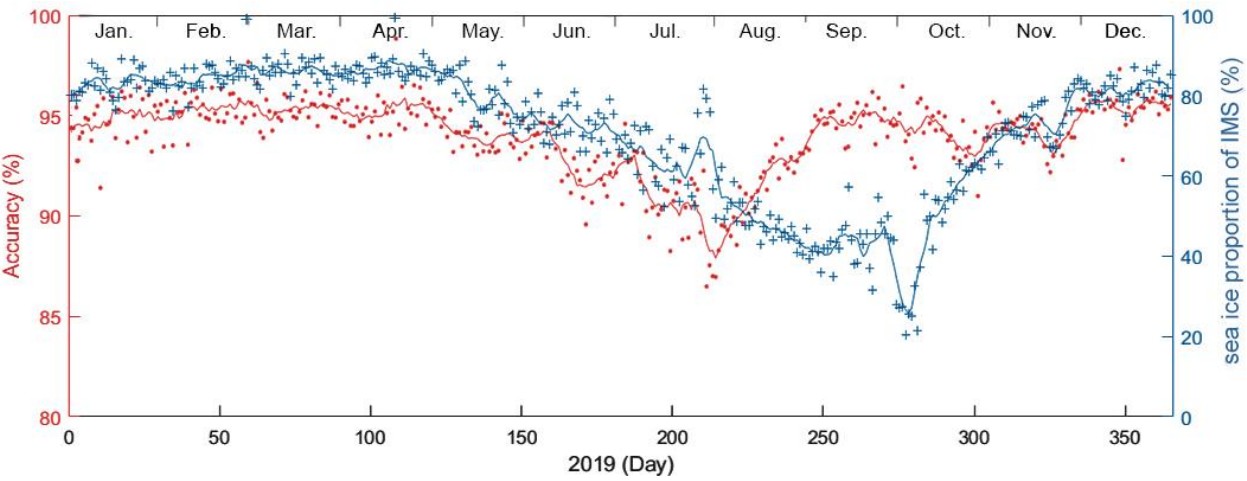

**Figure 13. Comparison between the S1-derived Arctic sea ice cover product and the IMS data for the whole year of 2019. The red
dots represent the daily average accuracy, and the red line is the 7-day average accuracy. The blue crosses signify the daily sea ice
proportion of the S1-covered region calculated based on the IMS data, and the blue line is the 7-day average sea ice proportion.**



## 5 Data availability

To facilitate related studies in the research community, the S1-derived sea ice data based on more than 28,000 images are organized into an Arctic sea ice cover product in a standard format. The S1-derived Arctic sea ice cover product is stored in NetCDF-3 and follows the Climate and Forecast Metadata CF-1.7 convention (Eaton et al., 2003) to allow easy access from a

range of standard tools across the leading computing platforms. Each record corresponds to the S1 images from which the Arctic sea ice cover product is derived. The product is named by SIC_ID_V1.0.nc, where SIC denotes sea ice cover, ID is the product ID of the S1 data, and V1.0 is the product version. Each record consists of 4 variables, namely, Longitude, Latitude, SeaIce, and Mask. The descriptions of the variables are listed in Table 1. Figure 14 shows an example of the Arctic sea ice cover product over a critical area, namely, the Northeast Passage, approximately located between 70° and 82° N, 40° and 80°

E. Taking full advantage of SAR data, this sea ice cover product has high spatial and temporal resolutions and offers satisfactory coverage.

The developed Arctic sea ice cover data by S1 SAR data in 2019 are available at: http://www.dx.doi.org/10.11922/sciencedb.00273 (Wang and Li, 2020). More S1 SAR data are being processed since its launch and the corresponding sea ice cover data in the Arctic will be also added to the repository.


**Table 1. List of variables and their descriptions in the NetCDF product.**

| No. | Variables | Descriptions |
|-----|-----------|--------------|
| 1 | Longitude | Longitude of each sea ice and land mask record |
| 2 | Latitude | Latitude of each sea ice and land mask record |
| 3 | SeaIce | 0 denotes open water, and 1 denotes sea ice |
| 4 | Mask | 0 indicates no land, and 1 indicates land |

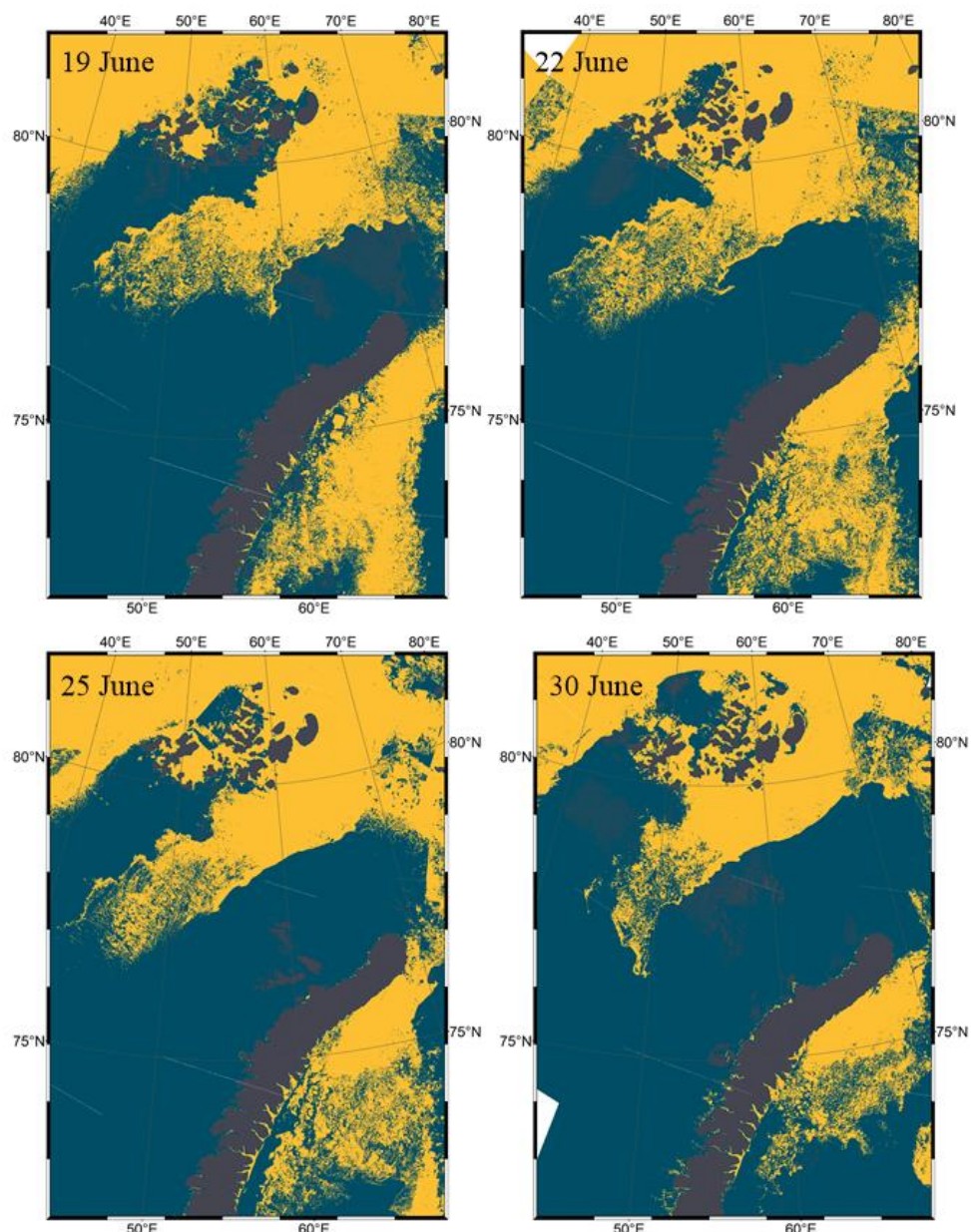

**Figure 14. Example of the Arctic sea ice cover product in the Barents Sea and Kara Sea of the Northeast Passage. Yellow represents sea ice, cyan represents open water, and gray represents land.**

**6 Discussion**

In late July and early August, the S1-derived sea ice information shows a higher discrepancy with the AMSR2 sea ice concentration data and IMS data compared to the rest of the year. To further investigate this phenomenon, the discrepancies



of the sea ice information derived from each scene of S1 data compared with the corresponding IMS and AMSR data are plotted in Figure 15. The dots represent the center coordinates of the S1 EW images, and their colors represent the values of

the absolute difference between the S1-derived sea ice concentration and the AMSR2 data (Figure 15 (a)) and the accuracy between the S1-derived results and IMS data (Figure 15 (b)). The time period of the comparison ranges from 25 July to 5 August 2019. Large discrepancies between the S1-derived results and the AMSR2 or IMS data are concentrated mainly in the East Siberian Sea and surrounding Greenland. During July and August, these two areas are mainly composed of extensive melting ice and large areas of brash ice. These complex ice conditions increase the difficulties of sea ice estimation by all

methods. In the following, we will present two examples in the East Siberian Sea and on the Greenland coast for further demonstration.

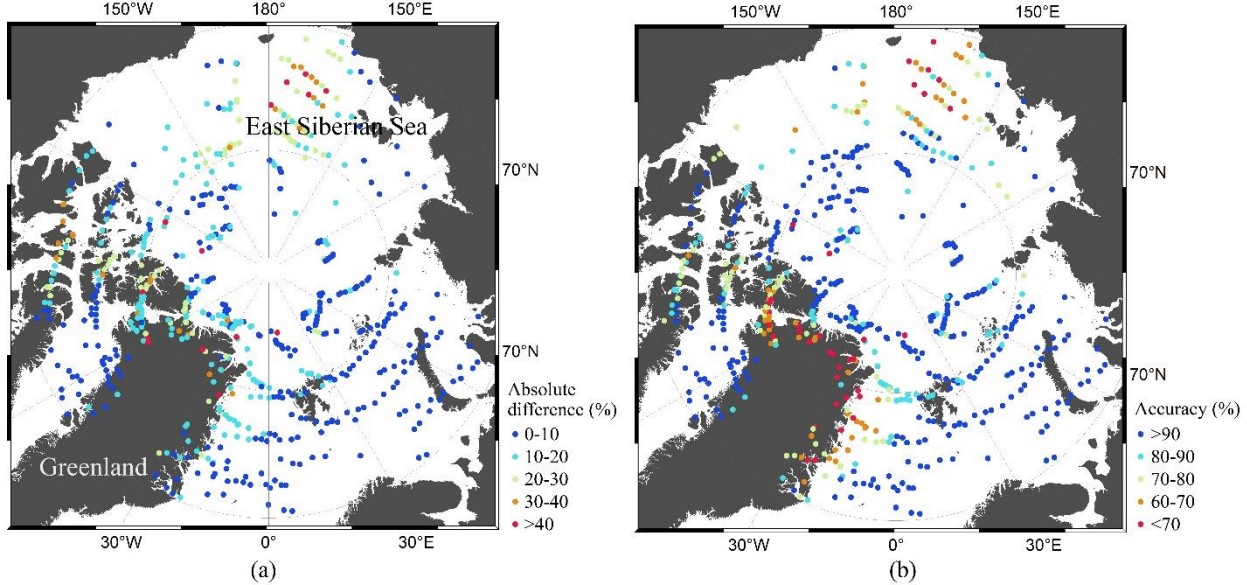

**Figure 15. Plots of comparisons between the S1-derived sea ice cover and other data in the period from 25 July to 5 August 2019. (a) Absolute difference between the S1-derived sea ice concentration and the AMSR2 data. (b) Accuracy between the S1-derived sea ice**
**cover results and the IMS data.**

Figure 16 shows an example of an image acquired on 29 July at 163° E, 74° N in the East Siberian Sea to illustrate the large discrepancies among the different data sources. Figure 16 (a) presents the RGB false-color composite image, and panel (b) depicts the corresponding S1-derived sea ice cover result. The southeastern area of the S1-derived results shows that brash ice is overestimated. Figure 16 (c) shows the IMS sea ice cover, which fails to detect the mixture of sea ice and open water and

instead reveals continuous sea ice cover. Figure 16 (d) shows the differences in the sea ice cover between the IMS and S1-derived results; the overall accuracy is 25.22 % for this example. Figure 16 (e) and (f) shows the S1-derived sea ice concentration and the AMSR2 data, while (g) shows the differences between them. Overestimation in the southeastern area of the S1-derived results leads to a high sea ice concentration, while the AMSR2 data show a much lower sea ice concentration than the S1-derived result, which causes a high average absolute difference of 28.06 %.

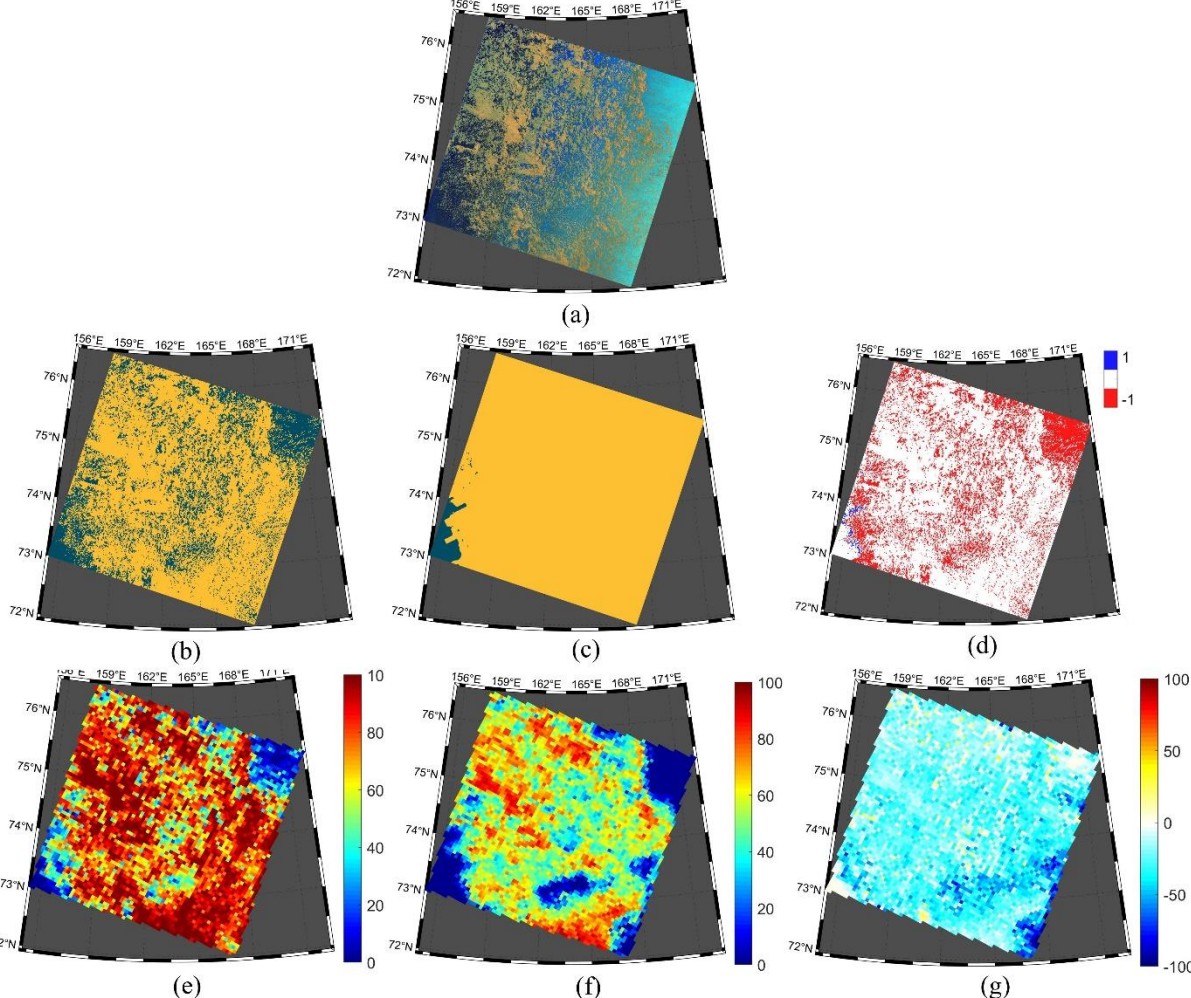

**Figure 16. Example of the S1-derived sea ice cover acquired on 29 July and its comparison with the AMSR2 and IMS data. (a) RGB false-color composite image. (b) and (c) S1-derived sea ice cover and IMS sea ice cover, respectively. (d) Differences in the sea ice cover between the IMS and S1-derived results (IMS − S1). (e) and (f) S1-derived sea ice concentration and the AMSR2 data, respectively. (g) Differences in the sea ice concentration between the AMSR2 data and S1-derived results (AMSR2 − S1). (Image ID: S1B_EW_GRDM_1SDH_20190729T200651_20190729T200751_017353_020A1F_94EB)**

Figure 17 is the same as Figure 16 but for the image acquired at 11° W, 77° N, near northeastern Greenland. In this example, the S1-derived result preserves the details of sea ice cover in the MIZ, but the IMS data still present the mixture of sea ice and open water as continuous ice, leading to an obvious overestimation and low overall accuracy of 61.56 % (Figure 17 (d)). The relatively large absolute difference of 13.97 % (Figure 17 (g)) between the S1-derived sea ice concentration and the AMSR2 data is due mainly to the AMSR2 data underestimating the pack ice in the northwest and slightly overestimating the brash ice in the northeast.

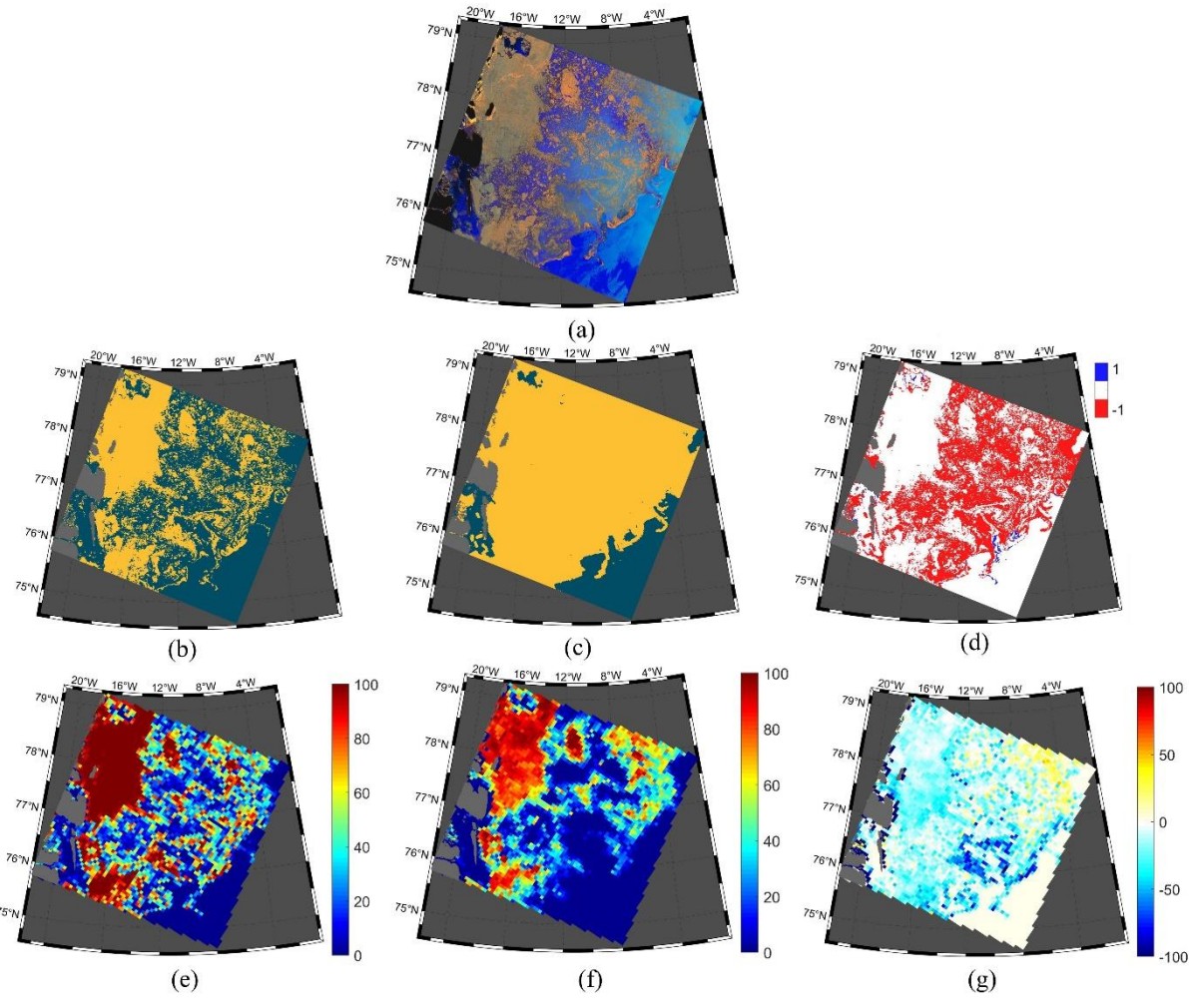

**Figure 17. Same as Figure 16 but for the photo acquired near Greenland. (Image ID: S1B_EW_GRDM_1SDH_20190802T080158_20190802T080258_017404_020BAA_48A5)**

We found that the IMS data tend to overestimate sea ice cover by recording brash ice as a whole ice surface, which forfeits

detailed features. This is likely because these data are obtained by implementing a daily average and smoothing the results

based on various observations acquired at different times (as well as at different spatial resolutions) within a day. Furthermore,

melted sea ice drifts significantly in the open sea during the summer season and leads to misclassification. We also found that

the AMSR2 sea ice concentration data tend to underestimate pack sea ice near the coast in the summer season. We inferred

that the changed radiometric characteristics of melted sea ice, particularly spatially extensive pack sea ice close to land,

interfere with the radiometer-based retrieval of sea ice concentration. Thus, during July and August, the aggregation of brash

ice and melted sea ice drifting along the coast of Greenland and within the East Siberian Sea causes high discrepancies among

the S1-derived results and the AMSR2 and IMS data. In addition, because land occupies part of the S1 data, there are fewer





valid areas involved in calculating the average discrepancy, which enlarges the proportion of local errors. This can also explain
why the discrepancy increases with the increasing proportion of land in the S1 images.

In the process of deriving sea ice cover from the S1 data, we also found some examples of distinct misclassification, specifically
for 208 cases in June, July, and August. The first typical type of such misclassification is caused mainly by a mismatch between
the S1-imaged land and the GSHHG land masking data. Figure 18 shows such an example acquired in northern Greenland.
Figure 18 (a) and (b) shows the S1 images in HV polarization without and with the land mask, respectively, clearly showing
that part of the land imaged by S1 is not fully masked by the GSHHG data. As described in Section 3.3, we stretched the HV-
polarized image by discarding 2 % of the maximum and minimum values each. However, the extremely high values of the
unmasked lands disturb the stretching process and cause information loss at low values, resulting in the misclassification results
shown in Figure 18(c); thus, to reduce the influence of the high radar backscatter caused by unmasked land, we changed the
threshold from 2 % to 5 %. Figure 18 (e) displays the corresponding correct classification result. After this modification for
all 208 cases, 156 scenes of data present the correct sea ice segmentation results; these scenes were added to the comparisons
with the IMS and AMSR2 data presented above.

Most of the remaining 52 cases (concentrated in the northern part of Greenland and Queen Elizabeth Islands; the S1B data
account for 42 of these cases) with distinct misclassifications show full sea ice cover except for land in the S1 images. Figure
19 shows two examples. The general impression of these two examples is that the sea ice radar backscatter exhibits significant
spatial variation, even resembling a "jump" in the HV-polarized data (e.g., the first example). The accumulated water on the
surface of melting sea ice can be inferred to trigger the dominant scattering mechanisms from volume scattering to surface
scattering and therefore reduces the radar backscatter in HV polarization. Consequently, these areas are misclassified as open
water. Notably, these 58 examples with distinct misclassifications were not included in the comparisons with the IMS and
AMSR2 data presented above.


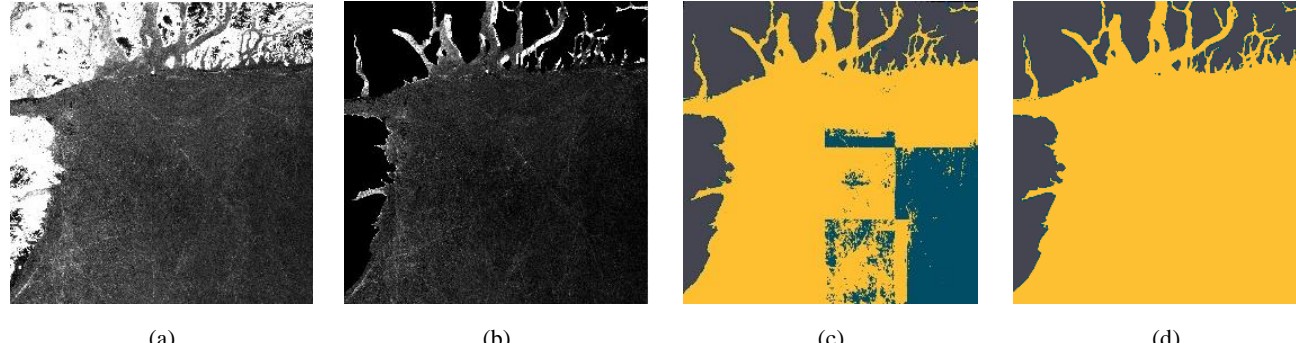

|         (a)         |         (b)         |         (c)         |         (d)         |

**Figure 18. Example of a misclassification due to a mismatch between the S1-imaged land and GSHHG mask. (a) HV-polarized S1
image with no land mask. (b) and (c) HV-polarized S1 images with the land mask, stretched with a threshold of 2 %, and the
corresponding misclassification result. (d) and (e) HV-polarized S1 images with the land mask, stretched with a threshold of 5 %,
and          the          corresponding          correct          result.          (Image          ID:**
**S1B_EW_GRDM_1SDH_20190701T185525_20190701T185629_016944_01FE25_A1F0)**





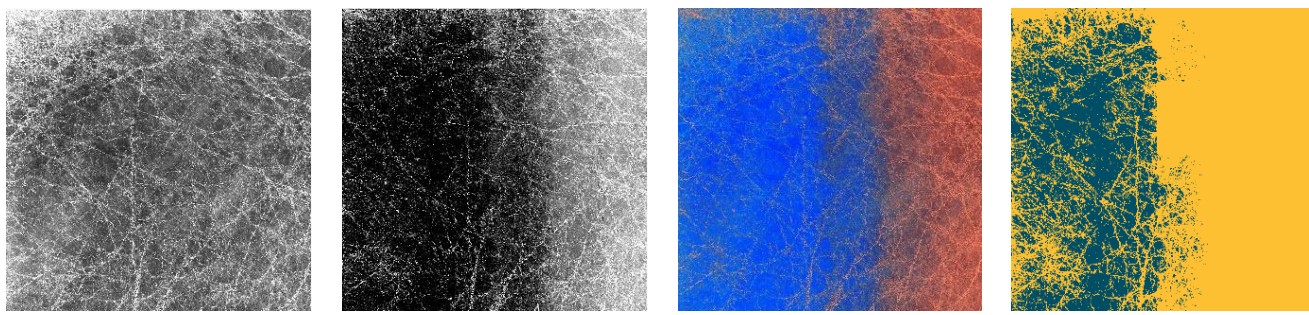

Image ID: S1B_EW_GRDM_1SDH_20190607T122033_20190607T122137_016590_01F3A2_2C14

Location: Northeast of Greenland. Center Longitude and Latitude: 38° E, 84° N

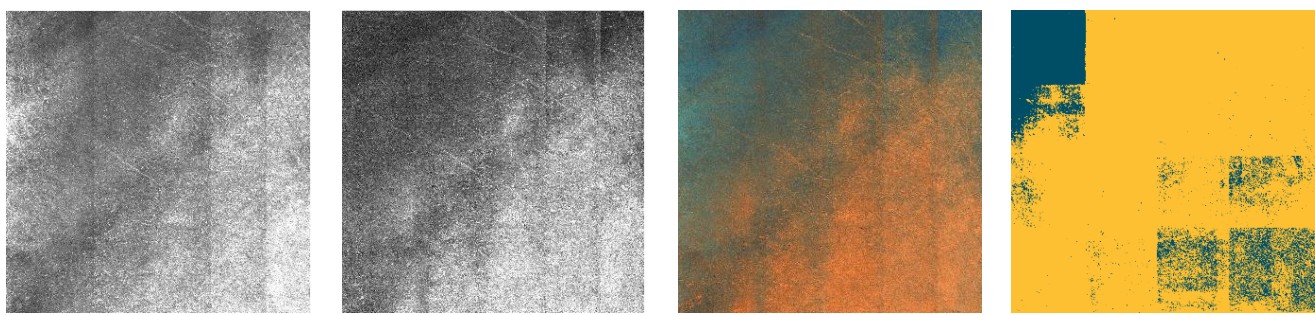

Image ID: SIE_S1B_EW_GRDM_1SDH_20190724T181612_20190724T181716_017279_0207FD_1D54

Location: North of the Queen Elizabeth Islands, Center Longitude and Latitude: 121° W, 85° N

**Figure 19. Examples of misclassifications that require further investigation. From left to right are S1 images in HH and HV polarization, RGB false-color composites and the S1-derived sea ice cover.**

## 7 Summary and conclusion

Due to the dramatic change of sea ice in polar regions, the most basic sea ice information, e.g., high-resolution sea ice cover
data, is drawing increasing attention for scientific research, polar navigation, and offshore operations. While commonly used
sea ice concentration data from spaceborne platforms are generally provided at spatial resolutions ranging from dozens of
kilometers to several kilometers, spaceborne SAR can provide such information at a spatial resolution of a few hundred meters
or even higher. Since the launch of S1A in 2014 and S1B in 2016, these two satellites have routinely acquired SAR data in
polar regions, and they are expected to continue operating for more than ten years, similar to ERS/SAR and ENVISAT/ASAR.
Therefore, S1-derived sea ice data with a high spatial resolution and high accuracy can offer great support for better
understanding the ongoing changes in polar regions. Thus, in this study, we aimed to develop a robust method of deriving sea
ice cover data in the Arctic by S1A and S1B, also attempted to generate easily accessed and handled product for other usage.
The proposed sea ice segmentation model is built upon the CNN architecture of U-Net, which can provide pixel-level
segmentation. When applying deep learning in the field of remote sensing, the lack of labeled data constitutes a major obstacle.
Thus, U-Net, which was designed for small training samples, is particularly suitable for the sea ice segmentation task. In



addition, the augmentation process dramatically increases both the quantity and the diversity of the training samples and further guarantees robust performance against overfitting even when only limited labeled data are available. Although HV-polarized SAR data are particularly suitable for sea ice detection, previous studies have suggested that combined information from both HH and HV channels can benefit this task. Therefore, we synthesized S1 HV-polarized data with polarization difference and

polarization ratio data between HH and HV polarization into an RGB false-color composite as the input to the U-Net model. Sea ice features in SAR images vary significantly. For instance, different types of sea ice can present variable radar backscatter characteristics, as can the same type of sea ice in different seasons. On the other hand, open water in the MIZ can also alert radar backscatter due to changes in the sea surface wind field. Therefore, it is difficult to train a single U-Net model to address various sea ice and open water states. Our solution is to train different U-Net models to obtain five classifiers with diverse

specializations. These five U-Net classifiers are combined by an integrated stacking model to generate an aggregate sea ice classifier, which enables every single model to fully utilize its strengths and mitigate its weaknesses, resulting in higher accuracy and sensitivity. Eventually, we applied the proposed model to more than 28,000 S1 images acquired in 2019, and consequently, we generated a sea ice cover product with a spatial resolution of 400 m.

We compared the S1-derived sea ice cover data obtained using the proposed model with AMSR2 sea ice concentration data

and IMS sea ice cover data. The average absolute difference between the S1-derived and AMSR2-derived sea ice concentrations is 5.55 %, and the overall accuracy of the S1-derived sea ice cover data is 93.98 % compared with the IMS data. Although this overall accuracy is promising, both comparisons reveal obvious seasonal fluctuations, particularly in the summer months of July and August. The melting of sea ice in summer is a major contributor to the large discrepancy between the S1 results and the AMSR2 data (noting that the IMS data also include information from radiometer data). Melting sea ice

not only has great impacts on microwave radiation, which consequently induces retrieval errors by radiometer data, but also alert radar backscatter as a result, the trained model may misclassify sea ice and open water. On the other hand, in the summer season, along with melting, thin sea ice tends to mix highly with open water, and this mixture may present radar backscatter characteristics similar to those of open water; thus, the model will overestimate the sea ice area. Nevertheless, the complex situation of sea ice in summer presents a challenge for deriving sea ice cover, and more efforts should be paid to this effect in

the future.

In this study, we conducted research constituting an important step in routinely generating SAR-derived sea ice cover products. More historical S1 images are being processed, and more comparisons with other datasets will be conducted. We hope that the developed high-resolution SAR-derived Arctic sea ice cover product can be utilized for different purposes by different communities and can further enhance scientific research, environmental protection, and resource utilization in the Arctic.

## 520 Author contributions

XML conceived the idea and designed the research. YRW developed the method and carried out the analysis. XML and YRW wrote the manuscript together.



**Competing interests**

The authors declare that they have no conflict of interest.

**Acknowledgment**

The S1 SAR data are downloaded from the Copernicus data hub (https://scihub.copernicus.eu/). The authors would like to thank the European Space Agency (ESA) for providing the S1 images to users worldwide. The use of the reference AMSR2 data (https://seaice.uni-bremen.de/start/), IMS data (https://www.natice.noaa.gov/ims/) and GSHHG data (https://www.soest.hawaii.edu/pwessel/gshhg/) is also acknowledged.

**Financial support**

The study is partially supported by the National Science Fund for Distinguished Young Scholars (42025605) and National Key Research and Development Project (2018YFC1407100) China.

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
