# Peer review of "Arctic sea ice cover data from spaceborne SAR by deep learning"

_Earth System Science Data, 2020_

## Referee Comment (RC1) · Anonymous Referee #1 · 3 Jan 2021

This paper presents an interesting exploration of using U-NET to segment Sentinel-1 data sets. The paper provided detailed methodology on the preprocessing of the image data, the construction of the model, and the training of the model. The resulting model has an impressive accuracy of 90%+.

I work mainly in machine learning, so my questions for this paper are about its methodology.

While I understand the difficulty of obtaining ground truth data for Arctic sea ice, the evaluation against lower resolution images is still strange. The S1 data set has a resolution of 400m, whereas the AMSR2 and IMS have resolutions in km. Validating the model against low-resolution data would potentially mask some of the errors of the model. It also limits the impact of the paper – as the paper seems to be motivated by

a need for high-resolution segmentation. What makes things worse was that the sea ice cover data from AMSR2 and IMS were also generated by other models, rather than real ground truth. This evaluation also shows a mismatch between the labels used for training and testing.

Can the authors validate the model against a higher resolution data set? Also, is it possible to validate using other S1 images with manual labels just as the authors did with the training data? In the paper, 1/3 of the 251 images were used as evaluation. It may be helpful to provide the evaluation results on those 1/3 images (I apologize if I missed it).

My other question is about the hyperparameters used in the paper, such as the "fixed thresholds of [2 dB, 7 dB] and [-2 dB, 3.5 dB]" and the batch size (there are quite a few more). How are these hyperparameter selected? Is there hyperparameter optimization?

A minor concern is the use of a small training data set (about 167 or 2/3 of 251). While U-Net was designed for a small training set, 167 seems to be too small, especially given that there are 28k images available. Will increase the training set or use a full CNN increase accuracy?

---

## Referee Comment (RC2) · Anonymous Referee #2 · 11 Jan 2021

This paper presents a method U-Net to classify Sentinel-1 SAR data into sea ice and open water (originally 40 m but changed to 400 m due to denoised operation) and validates their results based on lower resolution data (AMSR-2 of 6km and IMS of 1km) and provides the 2019 classification results for people to use. the overall intension of the paper seems good, but I have problem with their logical. First, in remote sensing field, nobody should use lower resolution data/results to validate results from high resolution data. I believe you should use similar or higher resolution images such as Sentinel-2 or other optical image to validate your results. Second, the paper used 251 images for labeling and for developing their methods, but there is no result about how good or bad of these exercises (usually you expect results about training accuracy and testing accuracy); the paper said used 5 classifiers but no mention about of what are

these five classifiers and which is better or best. The paper indeed mentioned about accuracy but it is about comparing with the entire year of AMSR data or IMS data, not from the labeling data (of 251 images). anyway, this is very confusing and it is misleading. Third, in terms of the five classifiers, the methods 2,3,5 (even 1 and 4), clearly have problems to separate windy water from ice, why you need to stacking them into your method; it also seems all the five methods have kind of problems to do so, but why once you stacking them together would result in a good separation of windy water from ice, overall? It seems questionable. Fourth, beside the windy water problem (confused with sea ice), it is a relatively easy task to classify sea ice and water from SAR images, I do not know why it is a big deal for this paper to develop such complex machine learning method to do so, while the validation of using low resolution data seems useless and illogical in my view. Fifth, the paper reduced the original 40 m pixel to 400 m pixel due to denoising, while the results are not that obvious from the figure 3, except for the HV data, one of the big line (separation) in the left portion of the image is gone. However, it seems very questionable, since, if this big line (separation) can be removed, why other small ones are still there? Sixth, the big problem for classification of sea ice in SAR imagery is the type of sea ice, such as thin ice, thick ice, even melt ponds and open water in summer. I hope their machine learning method would really help to resolve these problems. Seventh, I would think their resulted sea ice concertation dataset of 2019 would be quite good but just do not see why this method would be better than any other regular image classification method.

Figure 4, not sure about which parts in (a) (b) are water or ice. Should label them.

Figure 5, why need to convert dbz to 0-255? Why for HH-HV, HH/HV, need to twice for such conversions?

---

## Author Comment (AC1) · 10 Feb 2021

**Response to Reviewer**

We would like to thank the reviewer for the detailed and helpful comments, suggestions, and careful checking. Comments are responded on point-by-point basis. The comments from the reviewer are in black while the authors' responses are in blue. Where we refer to line numbers, the revised version with changes highlighted in blue is applicable.

**Overall comment by the reviewer**: This paper presents an interesting exploration of using U-NET to segment Sentinel-1 datasets. The paper provided detailed methodology on the preprocessing of the image data, the construction of the model, and the training of the model. The resulting model has an impressive accuracy of 90%+. I work mainly in machine learning, so my questions for this paper are about its methodology.

**Response:**

Thank the reviewer for his/her positive comment on this study. The raised comments and questions are addressed below.

**Comments 1:** While I understand the difficulty of obtaining ground truth data for Arctic sea ice, the evaluation against lower resolution images is still strange. The S1 data set has a resolution of 400m, whereas the AMSR2 and IMS have resolutions in km. validating the model against low-resolution data would potentially mask some of the errors of the model. It also limits the impact of the paper–as the paper seems to be motivated by a need for high-resolution segmentation. What makes things worse was that the sea ice cover data from AMSR2 and IMS were also generated by other models, rather than real ground truth. This evaluation also shows a mismatch between the labels used for training and testing.

Can the authors validate the model against a higher resolution data set? Also, is it possible to validate using other S1 images with manual labels just as the authors did with the training data? In the paper, 1/3 of the 251 images were used as evaluation. It may be helpful to provide the evaluation results on those 1/3 images (I apologize if I missed it).

**Response 1:**

In our research, the Arctic sea ice cover product is generated based on more than 28,000 S1 images and we intend to evaluate the product case by case. However, there are no other available (at least, easily accessible) sea ice cover and concentration products that have comparable spatial resolution and time-space coverage to the S1 EW data, for the case-by-case comparison. So we dialed it back and evaluated our data by comparing with the AMSR2 sea ice concentration data and IMS sea ice cover data, which have a lower resolution but are both widely used for sea ice mapping in polar regions. According to the suggestion, we conducted a comparison between the S1-derived Arctic sea ice cover data and the pixel-level visual inspection results. We manually labeled 96 cases of objectively selected S1 data as ground truth, and the validation based on those 96 cases shows an overall accuracy of 96.10%. A section "4.1 Comparison with Visual Interpretation Results" has been added to the revised manuscript and please refer to this section for more details.

**Comments 2:** My other question is about the hyperparameters used in the paper, such as the "fixed thresholds of [2dB, 7dB] and [-2dB, 3.5dB] "and the batch size (there are quite a few more). How are these hyperparameters elected? Is there hyperparameter optimization?

**Response 2:**

A more detailed illustration of the parameter determination has been added in the revised manuscript.

For the "fixed thresholds of [2dB, 7dB] and [-2dB, 3.5dB]":

*"These thresholds ensure approximately 95% of the HH/HV and HH-HV values falling into the range, which were determined according to the statistics of more than 200 S1 EW images acquired under different scenarios of sea ice and open water." (page 9, line 202-204)*

For the "batch size",

*"In our case, the performance of the model improves with the increasing of the batch size. However, due to the limitation of the memory capacity (16G), the batch size of 8 is the maximum value the device can handle." (page 11, line 254-256)*

For the "patch size of 256 × 256 pixels":
*"The patch size of 256 × 256 pixels is an empirical value weighed between manageable model size and sufficient information one patch has." (page 10, line 224-225)*

For other unexplained hyperparameters of the U-Net model, we followed the original settings by the authors of U-Net (Ronneberger et al., 2015). There is no automatic hyperparameter optimization.

Reference:

Ronneberger, O., Fischer, P., and Brox, T.: U-net: Convolutional networks for biomedical image segmentation, International Conference on Medical image computing and computer-assisted intervention, 234-241, https://doi.org/10.1007/978-3-319-24574-4_28, 2015.

**Comments 3:** A minor concern is the use of a small training dataset (about 167 or 2/3 of 251). While U-Net was designed for a small training set, 167 seems to be too small, especially given that there are 28k images available. Will increase the training set or use a full CNN increase accuracy?

**Response 3:**

The size of a S1 image is about 1,000 × 1,000 pixels after down-sampling (from original size of about 10,000 by 10,000). But the inputs of U-Net are image patches with dimensions of 256 × 256 pixels. This means numbers of small patches can be extracted within one S1 image. After patches extracting and training sample augmentation, the total number of training and evaluation dataset reaches more than 8000, which is a reasonable number for the U-Net model. To make it more clear, we have further illustrated the number of datasets in our revised manuscript. Please refer to subsection

3.4 (page 10, line 229-230).

Moreover, the selection of the training samples considered both the data acquisition location and the data acquisition season. We believe they covered most of the complex ice and water conditions. Considering that pixel-level manual labeling is quite time-consuming, not cost-effective to add more training samples for the moment.

---

## Author Comment (AC2) · 10 Feb 2021

**Response to Reviewer**

We would like to thank the reviewer for the detailed and helpful comments, suggestions and careful checking. Comments are responded on point-by-point basis. The comments from the reviewer are in black while the authors' responses are in blue. Where we refer to line numbers, the revised version with changes highlighted in blue is applicable.

**Overall comment by the reviewer**: This paper presents a method U-Net to classify Sentinel-1 SAR data into sea ice and open water (originally 40 m but changed to 400 m due to denoised operation) and validates their results based on lower resolution data (AMSR-2 of 6km and IMS of 1km) and provides the 2019 classification results for people to use. The overall intension of the paper seems good, but I have problem with their logical.

By responding to the detailed comments raised by the reviewer in the following, we tried to explain the logic of the present study.

**Comment 1:** First, in remote sensing field, nobody should use lower resolution data/results to validate results from high resolution data. I believe you should use similar or higher resolution images such as Sentinel-2 or other optical image to validate your results.

**Response 1:**

So far, there are no other available (at least, easily accessible) sea ice cover/concentration products that have comparable (or higher) spatial resolution to the S1 EW data. The reviewer mentioned that one can try optical data to validate the SAR-derived sea ice cover data. This is one solution but not an optimal one. On one hand, the optical data (e.g., S2 data) are often (even significantly) affected by the cloud. On the other hand, even by finding temporal-spatial collocated cloud-free optical data, we still need to identify sea ice cover from the optical data by visual inspection. This is

again not possible to evaluate the S1-derived sea ice cover data case by case, considering that more than 28,000 S1 images were used to generate the sea ice cover product.

In the presented study, we evaluated our data by comparing them with the AMSR2 sea ice concentration data and IMS sea ice cover data, which have a lower resolution but are both widely used for sea ice mapping in polar regions, particularly the sea ice concentration data retrieved from microwave radiometer (e.g., AMSR2 in this study). Therefore, validation and evaluation of the developed SAR-based sea ice cover data using data in low resolution is not the best but a reasonable solution.

Nevertheless, according to the suggestion, we further conducted a comparison between the S1-derived Arctic sea ice cover data and the pixel-level visual inspection results. We manually labeled 96 cases of objectively selected S1 data as ground truth, which has the same resolution as the sea ice cover product. The validation based on those 96 cases shows an overall accuracy of 96.10%. A section "4.1 Comparison with Visual Inspection Results" has been added in the revised manuscript and please refer to this section for more details.

**Comment 2:** Second, the paper used 251 images for labeling and for developing their methods, but there is no result about how good or bad of these exercises (usually you expect results about training accuracy and testing accuracy) the paper said used 5 classifiers but no mention about of what are these five classifiers and which is better or best. The paper indeed mentioned about accuracy but it is about comparing with the entire year of AMSR data or IMS data, not from the labeling data (of 251 images). Anyway, this is very confusing and it is misleading.

**Response 2:**

The performance of the five U-Net classifiers was tested based on the 96 cases of manually labeled data (the validation dataset with high spatial resolution, as mentioned in Response 1). Table 1 lists accuracy of the five individual classifiers, which ranges from 82% to 94%. The individual U-Net classifier is only the intermediate step, and the accuracy is not the main consideration when selecting the model, but the specialization

(has high accuracy for newly formed sea ice/ high wind sea surfaces/ ice floes). The thing that matters is the combined model has higher accuracy of 96.10% (by comparing visual interpretation of the 96 cases) than any individual model. Considering the length of this paper, the evaluation of the 5 individual classifiers is not presented in the manuscript.

Table R1. Accuracy of the 5 selected U-Net classifiers evaluated by visual interpretation results of 96 cases.

|  | TP | FP | TN | FN | Accuracy |
| --- | --- | --- | --- | --- | --- |
| **Classifier 1** | 37.97% | 14.07% | 44.39% | 3.57% | 82.36% |
| **Classifier 2** | 51.60% | 0.44% | 37.65% | 10.31% | 89.26% |
| **Classifier 3** | 50.64% | 1.40% | 42.69% | 5.28% | 93.32% |
| **Classifier 4** | 48.78% | 3.26% | 45.95% | 2.01% | 94.73% |
| **Classifier 5** | 47.86% | 4.18% | 46.21% | 1.75% | 94.07% |
| **Average** | 47.37% | 4.67% | 43.38% | 4.58% | 90.75% |

The performance of each model diverges upon being fed a different set of training data (but all belong to the 251 labeled S1 EW images). We divided the training samples into groups and trained the U-Net model with different datasets, thereby generating several classifiers. A better or best classifier is not defined based on its overall accuracy, but empirically selected considering their specialization: "models 2 and 3 are specialized for large areas of sea ice, especially newly formed sea ice, whereas models 1 and 4 produce fewer wrong segmentation results for high wind sea surfaces, and models 3 and 5 deliver more details when ice floes are mixed with water". We then adopted the idea of stacked generalization to enable each model to fully utilize its strengths and mitigate its weaknesses.

**Comment 3:** Third, in terms of the five classifiers, the methods 2, 3, 5 (even 1 and 4), clearly have problems to separate windy water from ice, why you need to stacking them

into your method; it also seems all the five methods have kind of problems to do so, but why once you stacking them together would result in a good separation of windy water from ice, overall? It seems questionable.

**Response 3:**

It is hard to find a perfect single classifier for all scenarios of sea ice and open water. For instance, as you mentioned, classifiers 2, 3, 5, clearly have problems to separate windy water from ice. However, the classifier 2 and 3 are specialized for large areas of sea ice, especially newly formed sea ice, and the classifier 5 delivers more details when ice floes are mixed with water.

As no single classifier is perfect, the integrated stacking model here we adopted was to fully utilize its strengths and mitigate its weaknesses. The process of integrated stacking can be considered as several classifiers vote for the final segmentation. The classifier will have higher voting weights in their specialized scenarios. The scenarios are described by ice cover proportion to the full coverage of an S1 EW scene, which is one of the inputs of our integrated stacking model. For instance, for the scenarios with high ice proportion, models 2 and 3 will have higher voices. The weights allocation is realized by a neural network. As mentioned in response 2, the accuracy of the five models ranges from 82% to 94%. After the integrating, the overall accuracy increases to 96%, which suggests the effectiveness of the designed integrated stacking model. To clarify this point, we have further explained the integrated stacking model in the revised manuscript. Please refer to subsection 3.6. (Page 13, Line 294-298).

**Comment 4:** Fourth, beside the windy water problem (confused with sea ice), it is a relatively easy task to classify sea ice and water from SAR images, I do not know why it is a big deal for this paper to develop such complex machine learning method to do so, while the validation of using low resolution data seems useless and illogical in my view.

**Response 4:**

The task of classifying sea ice and water from SAR images is not as (relatively) easy as it seems to be. The difficulties not only lie in windy water problem. The smooth thin

ice surface, melting ice, and the complex situation at ice and water mixing areas also make the task challenging. Those situations are not rare but always the case. Some S1 data were acquired on 11, Feb 2019 as examples to better illustrate the complicated sea ice and open water situation (see Fig. R1). These cases are challenging for automatic methods, and even for manual interpretation. As for the validation, pixel-level comparison with visual inspection results has been conducted, and please refer to response 1.

[Figure]

**Fig. R1.** Examples of HH-polarized and HV-polarized S1 EW images presenting the challenges in sea ice segmentation, acquired on 11, Feb 2019.

**Comment 5:** Fifth, the paper reduced the original 40 m pixel to 400 m pixel due to denoising, while the results are not that obvious from the figure 3, except for the HV

data, one of the big lines (separation) in the left portion of the image is gone. However, it seems very questionable, since, if this big line (separation) can be removed, why other small ones are still there?

**Response 5:**

The S1 HH data are de-noised using the noise vectors provided in the standard S1 EW data in both the azimuth and range directions. However, the provided noise vectors in the S1 EW data are not sufficient for the HV data, which are significantly affected by the "scalloping" effect and the variations in radar backscatter from beams to beams (i.e., the big line that the reviewer mentioned). Therefore, we developed a robust denoised method for the S1 EW HV data (Sun and Li, 2020), which functions well for all the S1 EW data processed by different IPF versions. The significant variations in radar backscatter from sub-swath to sub-swath are generally well removed using the proposed denoising method, particularly for the most significant "line" between the first and second sub-swath (the first sub-swath refers to the one counting from SAR near range direction). Moreover, the "biggest" line between the first and second sub-swath has the most significant impact on sea ice classification. After denoising processing, there are still residual multiplicative noise existing, i.e., "small lines" observed in the denoised images. However, for the application of sea ice monitoring by S1 data in HV data polarization, we found the residual multiplicative noise will not affect the sea ice classification and segmentation (Li et al., 2020).

Reference:

Sun, Y. and Li, X.-M.: Denoising Sentinel-1 Extra-Wide Mode Cross-polarization Images over Sea Ice, IEEE Transactions on Geoscience and Remote Sensing, https://doi.org/10.1109/TGRS.2020.3005831, 2020.

Li, X.-M., Sun, Y. and Zhang, Q.: Extraction of Sea Ice Cover by Sentinel-1 SAR Based on Support Vector Machine With Unsupervised Generation of Training Data, IEEE Transactions on Geoscience and Remote Sensing, 14-4, https://doi.org/10.1109/TGRS.2020.3007789, 2020.

**Comments 6:** Sixth, the big problem for classification of sea ice in SAR imagery is the type of sea ice, such as thin ice, thick ice, even melt ponds and open water in summer. I hope their machine learning method would really help to resolve these problems.

**Response 6:**

The reviewer is correct. We have been working on sea ice classification based on machine learning. The high spatial resolution of sea ice cover data by SAR is the first step of the full research plan.

**Comments 7:** Seventh, I would think their resulted sea ice concertation dataset of 2019 would be quite good but just do not see why this method would be better than any other regular image classification method.

**Response 7:**

Regular image classification methods have been implemented for sea ice classification on SAR images by previous research. Hong and Yang (2018) adopted the SVM approach for automatic sea ice discrimination based on the S1 EW data showing an accuracy of 92.1%-93.4%. However, noting that they cannot solve the noise problem that existed in the HV data and therefore, they have to exclude the first swath (it has a significant impact on sea ice classification and segmentation, as we mentioned in response 5) from a full S1 EW scene. We also adopted the SVM approach for sea ice segmentation (Li et al., 2020) after we solved the denoising problem of the S1 HV data. Moreover, our approach based on the SVM does not limit to fix training data. However, it is found that the SVM has a few problems with sea ice segmentation, e.g., discriminating between thin ice and windy open water. In this research, the developed method is robust for all kinds of scenarios of sea ice and open water, and has good accuracy of 96% (comparing with pixel-level ground truth). Moreover, our motivation is to process all the acquired S1 data to sea ice cover data in the Arctic for public sharing. Therefore, this study is not only method-oriented, but also focusing on generating a scientific dataset. As far as we know, there are no published SAR-based sea ice classification/segmentation methods that have been applied to a large amount of data as we did.

Reference:

Hong, D.-B., and Yang, C.-S.: Automatic discrimination approach of sea ice in the Arctic Ocean using Sentinel-1 Extra Wide Swath dual-polarized SAR data, International journal of remote sensing, 39, 4469-4483, https://doi.org/10.1080/01431161.2017.1415486, 2018.

**Comment 8:** Figure 4, not sure about which parts in (a) (b) are water or ice. Should label them.

**Response 8:**

According to this advice, the labels are added to Figure 4.

[Figure]

*Figure 1. Examples of S1 EW images presenting the challenges in sea ice segmentation. (a) HH-polarized and (b) HV-polarized S1 EW images of a windy sea surface. (c) HH-polarized and (d) HV-polarized S1 EW images of a smooth thin ice surface. (Image ID: (a) and (b) S1A_EW_GRDM_1SDH_20190130T060740_20190130T060840_025703_02DB20_85BC; (c and (d) S1B_EW_GRDM_1SDH_20190115T194015_20190115T194115_014509_01B066_11F1)*

**Comment 9:** Figure 5, why need to convert dbz to 0-255? Why for HH-HV, HH/HV, need to twice for such conversions?

**Response 9:**

The value range of the ordinary image data format is 0-255 (8-bit image). Thus, the liner stretch to 0-255 can simplify the process of data management and presentation. The polarization ratio (HH/HV) and polarization difference (HH-HV) are two parameters that proved to be useful for sea-ice discrimination and other feature extraction from SAR data (Dong et. al., 2012; Murashkin et. al., 2018). HH-HV presents the absolute differences between HH and HV polarization while HH/HV highlights the differences for small values, e.g., smooth sea surfaces with weak backscattered signals. The manuscript has been revised accordingly. Please refer to section 3.3.

Reference:

Murashkin, D., Spreen, G., Huntemann, M., and Dierking, W.: Method for detection of leads from Sentinel-1 SAR images, Annals of Glaciology, 59, 124-136, https://doi.org/10.1017/aog.2018.6, 2018.

Dong, J., Xiao, X., Sheldon, S., Biradar, C., Duong, N. D., Hazarika, M.: A comparison of forest cover maps in Mainland Southeast Asia from multiple sources: PALSAR, MERIS, MODIS and FRA. Remote Sensing of Environment, 127, 60-73, https://doi.org/10.1016/j.rse.2012.08.022, 2012.

**Overall response:**

The overall comment raised by the reviewer is that "The overall intention of the paper seems good, but I have problem with their logical". Therefore, after trying to address the raised questions and comments, we summarize the logic of this study below for an overall response:

So far, the widely used sea ice cover and concentration data in polar regions from remote sensing data are in relatively low spatial resolution. The high spatial resolution data from spaceborne SAR data are highly desirable. However, the challenge is that the presented sea ice and open water conditions in SAR images are very complicated.

Therefore, we chose to use the deep learning method to segment sea ice and open water. Moreover, after trying many times, we found it is not possible to use a single U-net classifier to deal with all scenarios of sea ice and open water. Then we decided to employ an integrated stacking model to combine multiple U-Net classifiers with diverse specializations, which is also explained in detail in response 3. To validate and evaluate the S1-derived sea ice cover data in the full year of 2019, as no other available dataset with a comparable spatial resolution with SAR, we have to conduct the comparison with AMSR2 sea ice concentration data and IMS sea ice cover data. Although it is not the best but a reasonable solution. Nevertheless, we further conducted a comparison between the S1-derived Arctic sea ice cover data and the pixel-level visual interpretation results based on 96 cases. All these validation and evaluation results suggest the promising performance of the proposed sea ice segmentation approach, with an overall accuracy of approximately 95%.

With the confidence of high accuracy of segmentation method, we developed a dataset using the full-year S1 data in 2019, and processing the S1 data acquired in other years is also ongoing. This is the full story of the study and its logic is clear.

---

## Author Response (AR2)

**Response to Review Comments**

**Arctic sea ice cover data from spaceborne SAR by deep learning**

**Report #1**

I would like to thank the authors for preparing the revision and addressing all of my concerns. This version of the paper is significantly improved. I do not notice any more major issues with this version. The only recommendations I have are about the presentation.

Thanks for the suggestions from the reviewer. The figures are updated and long sentences are modified for a better presentation. The changes to address the comments are discussed below and are highlighted in blue in the revised manuscript.

**Comments 1:** There are a few figures, whose fonts are too small to read, such as Figures 7, 9, and 11. Maybe the fonts can be increased for better readability.

**Response 1:**
The fronts have been increased for figures 6, 7, 9, 11, 15, 16, 17, and 18. Please refer to the revised manuscript.

**Comments 2:** Some of the sentences are long and hard to parse. And some sentences seem to become run-on sentences and comma splices. E.g. this sentence around line 320, "This image was acquired at the northeast of Severnaya Zemlya, presenting large areas of both open water and floating sea ice, as shown in Figure 10 (a), the RGB false-color composite image." There are a few similar sentences. I recommend checking all long sentences and break them into smaller ones if possible.

**Response 2:**
We have carefully read through the entire manuscript and made the necessary changes for the long sentences. For the mentioned sentence, it has been divided into two shorter ones as follows:
*"This image was acquired at the northeast of Severnaya Zemlya, presenting large areas of both open water and floating sea ice. Figure 10 (a) shows the RGB false-color composite image."*

**Comments 3:** Indenting the new paragraphs can improve readability.

**Response 3:**
The original format of the paragraphs was based on the template provided by ESSD. According to this advice, we use indentations for the first line of each paragraph in the revised manuscript.

**Report #2**

Authors addressed my comments well. I am ok to recommend the paper for publication after minor comments/suggestions been addressed.

Thanks for the comments and suggestions from the reviewer. The changes that we have made to address the comments are discussed below and are highlighted in blue in the revised manuscript.

**Comments 1:** in sections 2.2 and 2.3, you talked about pixel by pixel comparison between S1 and AMSR2, S1 and IMS, I wondering if you can really match them pixel by pixel (no overlapping)? If yes, how you did that to make them real match..,

**Response 1**:
The pixel-level match refers to the pixels of lower resolution data. The comparisons are further explained in sections 4.2 and 4.3.

For comparison between S1-derived results and AMSR2:
*"The S1-derived sea ice cover data (with a spatial resolution of 400 m) were converted into sea ice concentration on a regular grid of 6.25 km (the same as the spatial resolution of the AMSR2 data), with the center of each grid corresponding to each pixel of AMSR2. Then, the sea ice concentration data were matched with the AMSR2 data on a pixel-by-pixel basis."*

For comparison between S1-derived resutls and IMS:
*"The 400 m pixel size of the S1-derived sea ice cover data is comparable to that of the IMS data on a grid size of 1 km. We directly compared the IMS data and S1-derived results by matching the nearest pixels of the two data."*

**Comments 2:** for figure 4, a,b, there are dark areas, what they are? For c/d, is the lower right corner sea ice or water?

**Response 2**:
For figure 4 (a) and (b), the upper dark areas are land covered by the black land mask; while the dark areas at the lower right corner are open water with low radar backscatter intensity. For figure 4 (c) and (d), the lower right areas are sea ice. To clarify these two points, figure 4 has been updated with more labels.

[Figure]

*Figure 1. Examples of S1 EW images presenting the challenges in sea ice segmentation. (a) HH-polarized and (b) HV-polarized S1 EW images of a windy sea surface. (c) HH-polarized and (d) HV-polarized S1 EW images of a smooth thin ice surface. (Image ID: (a) and (b) S1A_EW_GRDM_1SDH_20190130T060740_20190130T060840_025703_02DB20_85BC; (c and d) S1B_EW_GRDM_1SDH_20190115T194015_20190115T194115_014509_01B066_11F1)*

**Comments 3:** figure 9, the last column, there is a gray/dark color type, beside the yellow and cyan, what is it? The same comment for the Figure 10 and figure 11;

**Response 3:**
The dark color in the RGB false-color composites and the gray color in the segmentation results both indicate the land mask. To clarify this point, the titles of figures 9, 10, 11, and 15 have been updated. Besides, the related description in section 3.2 has been revised as follows:
*"The land area is masked as black in the images by the Global Self-consistent Hierarchical High-resolution Geography Database (GSHHG, https://www.ngdc.noaa.gov/mg g/shorelines/gshhs.html) with the full grid resolution of 1×1 arc-minute."*

**Comments 4:** figure 12, I would suggest to use S1 concentration not the AMSR2 concentration, maybe including ASMR2 7-day average line for comparison, also the absolute difference, is S1-AMSR2, or AMSR2 – S1? Similar for the figure 14, SI concentration not IMS concentration, but adding its 7-day average….

**Response 4**:

According to this advice, the 7-day sea ice concentration/proportion based on S1-derived results has been added in figure 12 and figure 14. The daily plots of sea ice concentration/ proportion have been removed for better presentation. As for the absolute difference, S1-AMSR2, or AMSR2 – S1 makes no difference.

[Figure]

*Figure 2. Comparison between the S1-derived Arctic sea ice concentration data and AMSR2 data for the whole year of 2019. The red dots reflect the absolute daily difference, and the red line is the 7-day average absolute difference. The blue line is the 7-day average sea ice concentrations in the S1-covered area calculated based on the AMSR2 data, and the orange line is the 7-day average sea ice concentration calculated based on S1-derived results.*

[Figure]

*Figure 3. Comparison between the S1-derived Arctic sea ice cover product and the IMS data for the whole year of 2019. The red dots represent the daily average accuracy, and the red line is the 7-day average accuracy. The blue line is the 7-day average sea ice proportion in the S1-covered area calculated based on the AMSR2 data, and the orange line is the 7-day average sea ice proportion calculated based on S1-derived results.*